# Variation-aware Flexible 3D Gaussian Editing

**Hao Qin**[1,*] **Yukai Sun**[1,*] **Meng Wang**[1], **Ming Kong**[1,2,†] **Mengxu Lu**[1] **Qiang Zhu**[1,†]

[1]Zhejiang University
[2]Zhejiang Key Laboratory of Geographic Information Science
{haoqin,3220101205,22451130,zjukongming,lumengxu,zhuq}@zju.edu.cn

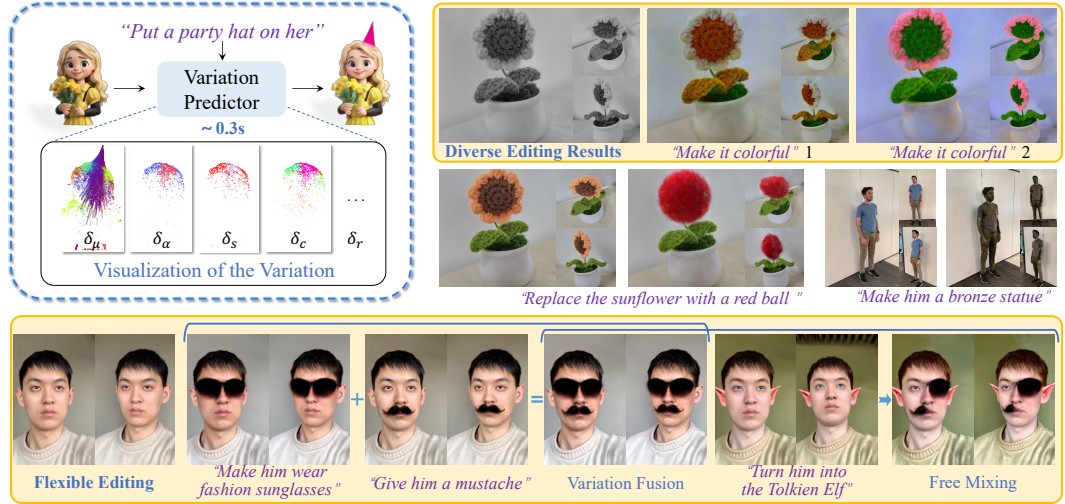

Figure 1: VF-Editor is a native editing method for 3D Gaussian Splatting across multiple scenes and instructions. In the top-left corner, we present a 2D visualization of the 3D variation within VF-Editor; please refer to App. C for specific visualization rules.

## ABSTRACT

Indirect editing methods for 3D Gaussian Splatting (3DGS) have recently witnessed significant advancements. These approaches operate by first applying edits in the rendered 2D space and subsequently projecting the modifications back into 3D. *However, this paradigm inevitably introduces cross-view inconsistencies and constrains both the flexibility and efficiency of the editing process.* To address these challenges, we present **VF-Editor**, which enables native editing of Gaussian primitives by predicting attribute variations in a feedforward manner. To accurately and efficiently estimate these variations, we design a novel variation predictor distilled from 2D editing knowledge. The predictor encodes the input to generate a variation field and employs two learnable, parallel decoding functions to iteratively infer attribute changes for each 3D Gaussian. Thanks to its unified design, VF-Editor can seamlessly distill editing knowledge from diverse 2D editors and strategies into a single predictor, allowing for flexible and effective knowledge transfer into the 3D domain. Extensive experiments on both public and private datasets reveal the inherent limitations of indirect editing pipelines and validate the effectiveness and flexibility of our approach.

## 1 INTRODUCTION

Personalized 3D editing is important in many fields such as virtual reality, industrial design, and game development, as it can substantially enhance creative efficiency and quality Botsch & Kobbelt (2005). With advancements in 2D-AIGC technology Saharia et al. (2022); Ramesh et al. (2021); Rombach

---

[*]Equal contribution
[†]Corresponding author

et al. (2022b) and 3D representation methods Mildenhall et al. (2021); Wynn & Turmukhambetov (2023); Kerbl et al. (2023); Chung et al. (2024), significant progress has been made in text-based 3D editing techniques Lu et al. (2024b); Mikaeili et al. (2023); Fang et al. (2024b); Chen et al. (2024b); Fang et al. (2024a).

A common strategy Haque et al. (2023); Vachha & Haque (2024) involves obtaining edited images from various views of a scene via 2D editors, then reconstructing the 3D scene based on these images. Although this approach enables diverse 3D edits, it still faces several challenges: 1) Since the 2D editor cannot ensure consistent editing patterns across views, *conflicts often arise between different views* in the reconstructed result. 2) The separate 2D editing and 3D reconstruction processes across different editing rounds *constrain both the flexibility and efficiency* of the final 3D editing outcomes. Some studies alleviate view inconsistencies by exchanging attention maps among views during 2D editing Dong & Wang (2023a); Hu et al. (2024); Karim et al. (2024); Wang et al. (2024a); Chen et al. (2024a). However, due to the black-box nature of neural networks, such approaches are insufficient to fundamentally resolve the inconsistencies across views. Moreover, research on enabling flexible interaction between different rounds of 3D editing remains relatively scarce.

In the field of 3D generation, increasing attention has been directed toward native generation based on feed-forward networks Li et al. (2025b); Wang et al. (2024c); Zhao et al. (2025b); Hong et al. (2024). This is because native 3D generative models fundamentally avoid the view inconsistency issues commonly associated with reconstruction-based approaches, while also significantly enhancing the flexibility of 3D content creation Xiang et al. (2025); Shi et al. (2025). Motivated by this, we aim to address the limitations of current indirect editing paradigms by training a native 3D editor. However, due to the scarcity of training data, it is infeasible to efficiently train such a feed-forward 3D editor using standard supervised learning techniques.

To this end, we propose an innovative framework, VF-Editor, which distills 2D editing priors into 3D editing knowledge and enables the training of a feed-forward 3D editor. Rich 2D editing priors provide sufficient knowledge for model training, but we find that a 3DGS editor that directly predicts the edited results is still difficult to converge. Given the explicit nature of 3D Gaussian Splatting Kerbl et al. (2023), if we can predict the variations of all primitives under a given editing instruction, the final edited result can be obtained by superimposing these variations onto the original attributes Lu et al. (2024a). Compared with directly predicting the edited outputs, modeling the variations alleviates the learning burden. Moreover, assigning precise variation values to each attribute of every primitive allows fine-grained control over the editing region and intensity, as well as the ability to compose multi-stage editing results for more personalized outcomes. Therefore, in VF-Editor, **we redefine the 3DGS editing task as a feed-forward variation prediction problem**.

The variation predictor of VF-Editor contains two key components: the variation field generation module and the parallel decoding function. The variation field generation module is used to encode input information and generate a variation field, while the parallel decoding function is employed to parallelly parse the variation of each 3D Gaussian from the variation field. By uniformly compressing specific variation quantities into the latent space, the variation field generation module effectively avoids the multi-round optimization process in traditional variation modeling methods Wu et al. (2024a); Fridovich-Keil et al. (2023). Additionally, the parallel decoding function achieves linear computational complexity related to the number of Gaussian primitives, and the decoding process can be accelerated through parallel computation. To mitigate the model convergence issues caused by the intercoupling of various non-structural attributes within the 3D Gaussians Charatan et al. (2024); Zou et al. (2024), we design two parallel decoding functions for iterative prediction of the variation.

The superiority of VF-Editor primarily lies in its ability to: **1)** distill multi-source 2D editing priors into a single model to meet various types of 3D editing requirements; **2)** accommodate inconsistencies across multiple views while enabling diverse inference; **3)** generalize effectively and support real-time editing for in-domain scenarios; **4)** offer enhanced interpretability and flexibility compared to traditional methods. We train and analyze VF-Editor on multi-source data, and the experimental results validate the effectiveness of our method. Our main contributions are:

- We propose VF-Editor, an innovative 3DGS editing framework that enables native editing in a feed-forward manner by distilling 2D editing priors into 3D space. VF-Editor not only effectively addresses the long-standing issue of multi-view inconsistency, but also significantly enhances the flexibility and efficiency of the editing process.

- We design a variation prodictor comprising a variation field generation module and two parallel decoding functions, achieving computational complexity linearly proportional to the number of Gaussian primitives. Moreover, it accommodates multi-source editing knowledge, effectively meeting diverse editing instructions.

- We evaluate our method under various settings and conduct comprehensive analyses; qualitative and quantitative results demonstrate its effectiveness and broad application potential.

## 2 RELATED WORK

**2D Editing** Substantial progress has been made in 2D editing techniques. IP2P Brooks et al. (2023) generates a dataset with GPT-3 Brown et al. (2020) and P2P Hertz et al. (2022) and fine-tunes StableDiffusion Rombach et al. (2022a) to perform text-guided editing. Subsequent works Zhang et al. (2023); Sheynin et al. (2024); Zhao et al. (2025a); Hui et al. (2024) achieve more flexible editing results by collecting higher-quality and more diverse datasets. Additionally, some works Hertz et al. (2022); Parmar et al. (2023); Tumanyan et al. (2023); Cao et al. (2023) explore using large-scale pre-trained text-to-image diffusion models to achieve text-guided image-to-image transitions. They first invert the input image into the noise space and then perform denoising guided by the instruction. GLIDE Nichol et al. (2022) trains a noised CLIP model to guide diffusion models, while Textual Inversion Gal et al. (2022) optimizes special text embeddings representing the target concept. DDIM Inversion Song et al. (2021); Mokady et al. (2022); Wallace et al. (2023) maps real images to noised latents with acceptable error. More recent studies Duan et al. (2023); Qian et al. (2024); Wu & la Torre (2023); Huberman-Spiegelglas et al. (2024) propose various techniques to mitigate the accumulated error in inversion. Different editing strategies excel at handling distinct types of editing instructions, each possessing unique visual editing knowledge. We attempt to utilize VF-Editor to store multiple 2D editing knowledge within a single model and construct 3D editing knowledge.

**3D Editing** Leveraging advanced 2D editing tools, numerous 3D editing methods have emerged. DreamEditor Zhuang et al. (2023) and Vox-E Sella et al. (2023) locate edit region through attention map and update the 3D data using Score Distillation Sampling (SDS) Poole et al. (2022). In contrast, Instruct-NeRF2NeRF Haque et al. (2023) integrates IP2P Brooks et al. (2023) to guide edits while preserving the overall structural integrity via iterative dataset updating. GenN2N Liu et al. (2024) introduces edit codes to differentiate between various editing styles, enabling diverse inference from a single instruction. ViCA-NeRF Dong & Wang (2023b) and DATENeRF Rojas et al. (2024) leverage depth information to enhance editing consistency across different views. More recently, numerous subsequent works Chen et al. (2024b); Fang et al. (2024a); Wu et al. (2024b); Wang et al. (2024b) achieve substantial improvements in editing efficiency by replacing NeRF with 3DGS. DGE Chen et al. (2024a) and Free-Editor Karim et al. (2024) jointly edit different views with epipolar-based feature injection, significantly enhancing the editing quality. However, the aforementioned methods require the conversion of 3D data into 2D images during the editing process, which leads to inconsistencies across different views that cannot be fundamentally resolved. Shap-Editor Chen et al. (2023) proposes a latent-space editing method for NeRF, eliminating the need for 2D data during inference, but it is only capable of handling simple objects and lacks flexibility. 3DSceneEditor Yan et al. (2024) achieves object addition, deletion, or relocation by segmenting individual objects from the scene, while GSS Saroha et al. (2024) realizes 3D stylization by modifying the color coefficients of each primitive. However, their support for only a single type of edit makes them inadequate for flexible editing. 3D-LATTE Parelli et al. (2025) and VoxHammer Li et al. (2025a) enable native 3D editing through the use of 3D diffusion models. However, their reliance on pretrained 3D generators inherently constrains the diversity of data distributions they can handle. In this paper, we aim to design a universal, flexible, and rapid editing tool for 3D Gaussians.

## 3 METHOD

VF-Editor first trains the variation predictor $\mathcal{P}_\theta$ by distilling 2D editing knowledge, after which it can perform real-time 3D editing across multiple instructions and scenes. We will first introduce the architecture of $\mathcal{P}_\theta$, then explain its training and inference processes. The overall schematic of VF-Editor is shown in Figure 2.

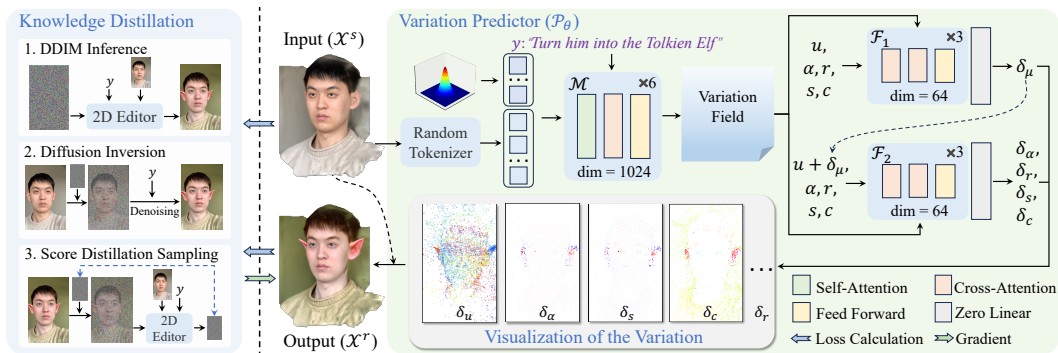

Figure 2: Schematic of VF-Editor. Given a 3D scene $\mathcal{X}^s$ and an editing instruction $y$, the variation predictor $\mathcal{P}_\theta$ generates variations which, when overlaid on the input scene $\mathcal{X}^s$, produce the edited result $\mathcal{X}^r$. VF-Editor trains $\mathcal{P}_\theta$ by distilling multi-source visual editing knowledge.

## 3.1 VARIATION PREDICTOR

Given the source 3D model $\mathcal{X}^s$ and an editing instruction $y$, $\mathcal{P}_\theta$ can predict the variations $\Delta = \{\delta_\mu, \delta_s, \delta_\alpha, \delta_c, \delta_r\}$. Each of $\mu$, $s$, $\alpha$, $c$, and $r$ denotes the mean, scale, opacity, color, and rotation of 3D Gaussians, respectively. The edit result $\mathcal{X}^r$ is obtained by overlaying $\Delta$ onto $\mathcal{X}^s$ :

$$\mathcal{P}_\theta : (\mathcal{X}^s, y, \varepsilon) \to \Delta, \quad \mathcal{X}^r = \mathcal{X}^s + \Delta, \tag{1}$$

where $\varepsilon \sim \mathcal{N}(0, I)$. $\mathcal{P}_\theta$ consists of a random tokenizer $\mathcal{T}$ and two key components: the variation field generation module $\mathcal{M}$ and a set of iterative parallel decoding functions $\mathcal{F}$. Specifically, $\mathcal{M}$ is used to integrate the features of $\mathcal{X}^s$ and $y$, and to generate the variation field, while $\mathcal{F}$ can rapidly extract $\Delta_i$ for each 3D Gaussian from the variation field.

**Random Tokenizer** ($\mathcal{T}$)    To handle different numbers of 3D Gaussians, we first design a random tokenizer that transforms the 3D Gaussians into a fixed number of tokens. Specifically, we randomly select $n$ 3D Gaussians from $\mathcal{X}^s$ as anchor points, with the remaining 3D Gaussians serving as data points. For each anchor point, we choose $k$-1 3D Gaussians from the data points that are spatially closest to form a group, thereby decomposing $\mathcal{X}^s$ into $n$ 3D tokens, each of dimensionality $k * f$, where $f$ is the dimension of a Gaussian primitive. To facilitate subsequent computations, an MLP is applied to each 3D token for dimensional transformation. Given the non-uniform spatial distribution of 3D Gaussians Feng et al. (2024b); Qu et al. (2025); Feng et al. (2024a), we adopt random sampling instead of the conventional farthest point sampling Qi et al. (2017); Zhao et al. (2021) commonly used in point cloud processing to select anchor points. This choice avoids the over-selection of sparse edge primitives, leading to a more reasonable distribution of sampled anchor points.

**Variation Field Generation Module** ($\mathcal{M}$)    We hypothesize that a key factor contributing to multi-view inconsistency in 3D editing is the inherent probabilistic flow of existing 2D editing methods. This flow leads to high variability in 2D editing results, making precise control challenging Cai & Li (2025). While restricting 2D editing diversity can significantly improve consistency across views, it comes at the cost of reduced diversity in the 3D editing results Chen & Wang (2024). To thoroughly avoid inconsistencies across views, we choose to store the possible outcome of 2D editing into $\mathcal{P}_\theta$, that is, **to preserve rather than limit the probabilistic flow during the distillation process**. Specifically, we preserve the *key noise* $\varepsilon$ that is strongly correlated with the probability flow (e.g., the initial noise in DDIM inference Song et al. (2021)), and concatenate it with the 3D tokens as inputs to $\mathcal{M}$. This distillation idea of using *key noise* to retain probability flow has been proven to be effective in the field of diffusion acceleration Kang et al. (2024); Yin et al. (2024); Lin et al. (2024). For details of the precise storage strategy of $\varepsilon$, please refer to Section 3.2.2. As shown in Figure 2, $\mathcal{M}$ is constructed by stacking transformer blocks Vaswani et al. (2017). The instruction $y$ is encoded by the CLIP text encoder Radford et al. (2021) and then injected into the 3D tokens through cross-attention layers to form the variation field:

$$f_\Delta = \mathcal{M}(\mathcal{T}(\mathcal{X}^s) \oplus \varepsilon; y), \tag{2}$$

where, $\oplus$ denotes the concatenation operation, and $f_\Delta$ represents the variation field.

**Iterative Parallel Decoding Function ($\mathcal{F}$)**   Unlike common strategies employed in dynamic scene reconstruction Wu et al. (2024a); Cao & Johnson (2023), we do not convert the variation field into a triplane. Instead, we design the parallel decoding function to decode the variation of each Gaussian primitive in parallel. Specifically, we employ a transformer architecture without self-attention to represent this decoding function, taking all attribute values of each 3D Gaussian as input (*query*) and the variation field as the condition (*key* and *value*). The variation of each 3D Gaussian is decoded independently, allowing parallel processing to accelerate computation. To alleviate the issue of 3D Gaussians tending to alter appearance rather than move positions, we design an iterative decoding strategy by separating the mean $\mu$ from other attributes:

$$[\delta_\mu] = \mathcal{F}_1(\mathcal{X}_\mu^s, \mathcal{X}_\alpha^s, \mathcal{X}_s^s, \mathcal{X}_c^s, \mathcal{X}_r^s; f_\Delta), \quad [\delta_s, \delta_\alpha, \delta_c, \delta_r] = \mathcal{F}_2(\mathcal{X}_\mu^s + \delta_\mu, \mathcal{X}_\alpha^s, \mathcal{X}_s^s, \mathcal{X}_c^s, \mathcal{X}_r^s; f_\Delta). \quad (3)$$

Given that the feature dimension of each 3D Gaussian is relatively low, the dimension of $\mathcal{F}$ can be correspondingly reduced, keeping the model lightweight. To stabilize the initial training process, we insert "zero linear" (a linear layer initialized with zeros) layers at the end of $\mathcal{F}$, ensuring the initial outputs of $\mathcal{P}_\theta$ are zero, thus providing more effective initial gradients for training.

## 3.2 Knowledge Distillation

VF-Editor trains $\mathcal{P}_\theta$ by distilling 2D editing knowledge. Since we have not introduced significant domain-specific designs into $\mathcal{P}_\theta$, it can theoretically store knowledge from multiple editing modalities:

$$\{\mathcal{E}_{T_1}, \mathcal{E}_{T_2}, ..., \mathcal{E}_{T_N}\} \xrightarrow{\text{Distill}} \mathcal{P}_\theta, \quad (4)$$

where, $\mathcal{E}_{T_i}$ denotes different 2D editing models or strategies. To achieve knowledge distillation from multiple sources, we employ datasets from diverse domains and utilize various editing models to generate a wide range of training samples. In the following, we list our collected data and then explain the different editing strategies applied.

### 3.2.1 3D Data

Traditional 3D editing methods commonly utilize a fixed set of 3D-instruction pairs for testing, but these pairs are insufficient to provide adequate training data for $\mathcal{P}_\theta$. So, we collect additional 3D data from various domains.

**Reconstructed Objects (RObj):** ShapeSplat Ma et al. (2024) contains 65K reconstructed objects from 87 unique categories, from which we select 662 high-quality 3D models across 32 categories as part of our training set. Most objects in ShapeSplat lack rich color variations; therefore, we primarily evaluate VF-Editor's capability for global style editing on this subset, such as instructions like "*make its color look like rainbow.*"

**Generated Objects (GObj):** In addition to the reconstructed objects, we enrich our training set by generating a batch of 3D objects through generative models. Specifically, we first employ SD3 Esser et al. (2024) to produce 500 cartoon character images with diverse appearances. These images are then fed into V3D Chen et al. (2024c) to obtain corresponding 3D objects, from which we select 319 high-quality generated 3D models. In this subset, we primarily evaluate the capability of VF-Editor in local detail editing tasks, such as "*put a party hat on him.*"

**Reconstructed Scenes (Scene):** We also conduct training on several 3D scenes, specifically including three public 3D scenes (`face`, `person_small`, `fangzhou_small`), three private 3D scenes (`doll_grayscale`, `sunflower_grayscale`, `sunflower`), and 115 generated street-view scenes Chung et al. (2023). Besides employing the editing instructions commonly utilized in previous works, we additionally train colorization, style transfer, and replacement instructions based on various 2D editing strategies for these scenes. Except for the generated scenes, all other scenes are reconstructed using Mini-splatting Fang & Wang (2024) to obtain the 3D Gaussians.

### 3.2.2 Editing Strategy

Different 2D editing models and editing strategies encapsulate distinct editing knowledge. We attempt to distill various types of such knowledge into $\mathcal{P}_\theta$ and construct 3D editing knowledge accordingly.

**DDIM Inference:** Using a 2D editor to perform DDIM inference for obtaining edited images is a widely adopted 2D editing strategy. For the 3D data in RObj and GObj, we utilize IP2P Brooks et al.

(2023) to edit the rendered images and store the {*initial noise*}–{*instruction*}–{*edited image*} triplets. The deterministic nature of the DDIM sampler ensures a one-to-one correspondence between the initial noise and the edited image Song et al. (2021). Therefore, we incorporate the initial noise as $\varepsilon$ to preserve the probabilistic flow in IP2P. Considering that IP2P is not particularly adept at coloring tasks and to further validate the ability of VF-Editor in distilling multi-model knowledge, we also employ CtrlColor Liang et al. (2024) for DDIM inference on the Scene dataset to collect triplets applicable to coloring tasks.

**Diffusion Inversion:** Benefiting from the excellent properties of diffusion models, 2D editing can be achieved using only a 2D generator. We adopt the DDPM inversion strategy proposed by Huberman-Spiegelglas et al. (2024) to edit images and collect triplets suitable for replacement tasks. Due to the uncertain nature of the DDPM sampler Ho et al. (2020), storing all noise in the trajectory to triplets is excessively redundant. To simplify computation, we retain only the noise sampled from the Gaussian distribution in the final step of inversion as $\varepsilon$. Although this approach does not ensure a one-to-one correspondence between $\varepsilon$ and the edited result, we find that due to the sparsity of the data, the model can still identify a degenerated probabilistic flow that leads to convergence.

Once sufficient triplets are collected through DDIM inference and diffusion inversion, we can proceed to train $\mathcal{P}_\theta$. $\mathcal{X}^s$ and $\varepsilon$ are used as inputs, $y$ is used as a conditioning signal, and the *edited image* serves as the target for the image rendered by the edited result $\mathcal{X}^r$:

$$\mathcal{L}_{\text{din}} = \mathbb{E}_{\mathcal{X}^s, y, \varepsilon} \left[ d(\mathcal{R}(\mathcal{P}_\theta(\mathcal{X}^s, y, \varepsilon) + \mathcal{X}^s), x^e) \right] = \mathbb{E}_{\mathcal{X}^r} \left[ d(\mathcal{R}(\mathcal{X}^r), x^e) \right], \quad (5)$$

where $\mathcal{R}$ represents differentiable rasterization rendering, and $x^e$ refer to the *edited image*. $d$ represents the distance metric function, and in VF-Editor, we use Mean Squared Error (MSE). For simplicity, we omit the camera parameters in Equation 5. The specific number of training data collected through DDIM inference and Diffusion inversion is shown in Table 1. Incor-

Table 1: The quantity of various training data.

| Type | 3D Data | Instruction | 3D-Instruction | Triplet |
|------|---------|-------------|----------------|---------|
| RObj | 662 | 4 | 2,490 | 18,355 |
| GObj | 319 | 9 | 847 | 9,261 |
| Scene | 121 | 7 | 126 | 4950 |
| All | 1102 | 20 | 3,463 | 32566 |

porating $\varepsilon$ into the input and supervising with only one view's editing result effectively mitigates the issue of multi-view inconsistency. Moreover, we observe that, given sufficient training data, $\mathcal{P}_\theta$ is capable of inferring changes in novel views based on variations observed in the known ones.

**Score Distillation Sampling (SDS):** Besides performing data distillation by collecting triplets, we also attempt to utilize SDS Poole et al. (2022) to distill knowledge from the 2D editor:

$$\mathcal{L}_{\text{sds}} = \mathbb{E}_{t, y, \mathcal{R}(\mathcal{X}^s), \varepsilon} \left[ w(t) \left( \varepsilon_\phi(z_t; t, y, \mathcal{R}(\mathcal{X}^s)) - \varepsilon \right) \right], \quad (6)$$

where $w(t)$ is a weighting function that depends on the timestep $t$, $\varepsilon_\phi$ is the noise predictor within the 2D editor Brooks et al. (2023), and $z_t$ is the feature vector obtained by adding $\varepsilon$ to latent embedding of $\mathcal{R}(\mathcal{X}^r)$. For $\mathcal{L}_{\text{sds}}$, triplets need not be collected offline, and the distillation process is unaffected by the quality of images generated by the 2D editor. However, since SDS only provides indirect validation rather than direct supervision, mode collapse often occurs in the editing results, leading to a loss of diversity Wang et al. (2023); Zhuo et al. (2024). Consequently, $\mathcal{L}_{\text{sds}}$ is not employed as the primary means of distillation. Nevertheless, the $\mathcal{P}_\theta$ trained using $\mathcal{L}_{\text{sds}}$ offers a robust baseline solution, with specific effects and discussions presented in Section 4.5 and 4.6.

### 3.3 INFERENCE

After distillation, $\mathcal{P}_\theta$ enables real-time editing for in-domain scenarios. Given the source 3D Gaussians and a noise sampled from the standard Gaussian distribution, along with an instruction, $\mathcal{P}_\theta$ generates the corresponding variations. The edited result can be obtained by applying the variations to the source 3D Gaussians, and the entire editing process takes approximately 0.3 seconds.

## 4 EXPERIMENTS

### 4.1 IMPLEMENTATION DETAILS

**Network Architecture:** We set $n$ and $k$ to 256 and 128, respectively, and map the dimension of each token to 4096 after MLP projection. Noise sampled from the standard Gaussian distribution,

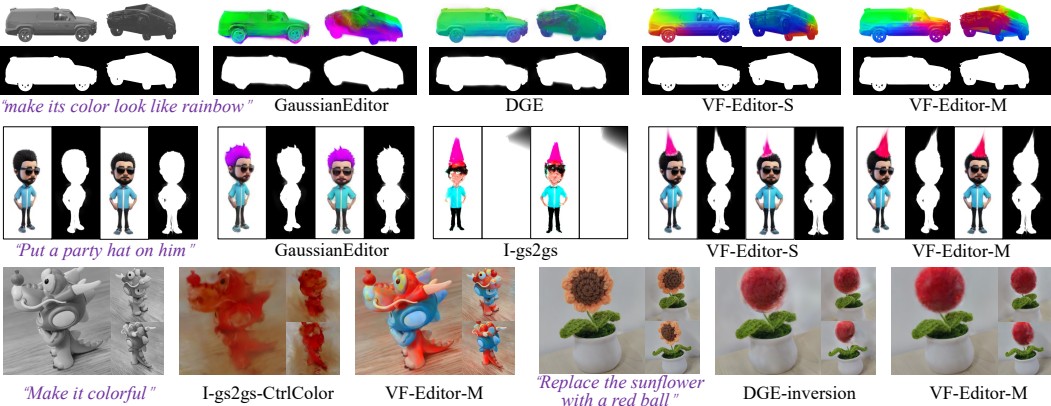

Figure 3: Qualitative comparison. VF-Editor achieves desired 3D editing with maximal preservation of original information. For video results, please see *Demo.mp4* in the supplementary materials.

Table 2: Comparison with other editing methods. VF-Editor achieves the best performance.

| Method | RObj | | | GObj | | | Scene | | | IAA↑ |
|---|---|---|---|---|---|---|---|---|---|---|
| | $IS\uparrow$ | $C_{sim}\uparrow$ | $C_{con}\uparrow$ | $IS\uparrow$ | $C_{sim}\uparrow$ | $C_{con}\uparrow$ | $IS\uparrow$ | $C_{sim}\uparrow$ | $C_{con}\uparrow$ | |
| I-gs2gs | 3.86 | 0.193 | 0.659 | 3.51 | 0.176 | 0.863 | 3.37 | 0.112 | 0.872 | 4.74 |
| GaussianEditor | 3.25 | 0.261 | 0.736 | 3.19 | 0.194 | 0.865 | 3.65 | 0.107 | 0.897 | 4.89 |
| DGE | 3.10 | 0.252 | 0.752 | 2.95 | 0.191 | 0.879 | 3.54 | 0.093 | 0.894 | 5.05 |
| VF-Editor-M | **4.32** | **0.296** | 0.763 | 4.15 | 0.206 | 0.875 | **4.06** | 0.127 | **0.903** | **5.24** |
| VF-Editor-S | 4.31 | 0.292 | **0.767** | **4.24** | **0.227** | **0.881** | 4.04 | **0.132** | 0.895 | 5.19 |

with a size of $1 * 4 * 64 * 64$, is reshaped and concatenated with 3D tokens. Editing instructions are encoded by CLIP and fed into the cross-attention layer of $\mathcal{M}$. We do not impose any constraints or regularization on 3D Gaussian attributes within $\mathcal{F}$ to ensure the flexibility of the generated variations.

**Training:** In Section 3.2.2, we introduce two optimization objectives, $\mathcal{L}_{din}$ and $\mathcal{L}_{sds}$, for $\mathcal{P}_\theta$. Regarding $\mathcal{L}_{din}$, the batch size is set to 16, and the entire training process takes 52 hours on 4 A100 GPUs. For $\mathcal{L}_{sds}$, the batch size is set to 8*4, and the training process takes 90 hours on 1 A100 GPU. Since $\mathcal{L}_{sds}$ tends to cause the output of $\mathcal{P}_\theta$ to collapse to a unique solution, we focus on $\mathcal{L}_{din}$ for our investigations in Section 4.2 and 4.3. To facilitate analysis, the $sh$ degree of 3D Gaussians is set to 0.

**Metrics:** Following Haque et al. (2023); Chen et al. (2024a), we compute the CLIP Text-Image Direction Similarity ($C_{sim}$) and CLIP Direction Consistency ($C_{con}$). Additionally, we calculate the Inception Score ($IS$) and conduct the Image Aesthetics Assessment ($IAA$) using Yi et al. (2023) to further evaluate the quality and diversity.

## 4.2 COMPARISON

We compare VF-Editor with three 3DGS editing methods, Instruct-gs2gs (I-gs2gs) Vachha & Haque (2024), GaussianEditor Chen et al. (2024b), and DGE Chen et al. (2024a). Given the domain gap in the 3D datasets we collected, we train two versions of VF-Editor: VF-Editor-S, trained on a single-domain dataset, and VF-Editor-M, trained on multi-domain datasets.

In Figure 3, we demonstrate visual comparisons with the baseline. Previous works on 3D editing typically utilize instructions with *abundant prior information* when evaluating their effects. However, as observed in the first row of Figure 3, these methods *nearly fail to function when faced with instructions lacking prior information, even when the editing instructions are very simple*. Regarding the coloring experiment, for a fair comparison, we replace IP2P Brooks et al. (2023) in I-gs2gs Vachha & Haque (2024) with CtrlColor Liang et al. (2024). It is apparent that the strategy of iteratively substituting 2D images is inadequate for the coloring task. In the replacement experiment, we employ the same attention injection strategy as in DGE Chen et al. (2024a) to enhance the denoising process during diffusion inversion. This allows us to obtain edited images from various views, which serve

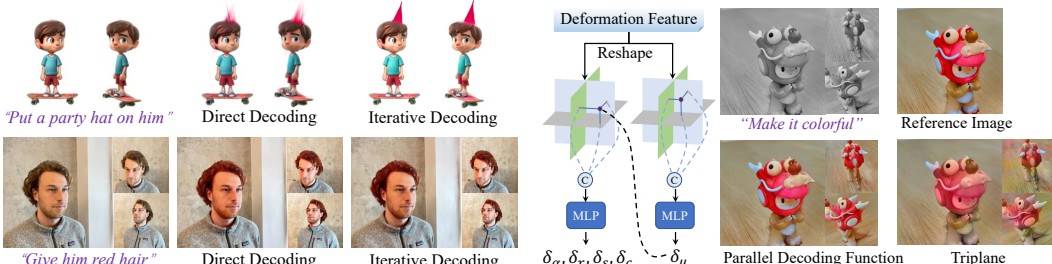

Figure 4: Visualization of the ablation study of iterative decoding. Direct decoding impairs the model's predictive capability regarding the positional changes of the 3D Gaussian.

Figure 5: Visualization of the ablation study of parallel decoding function. (Left) The triplane decoding strategy used for ablation. (Right) Display of the reference image and editing results.

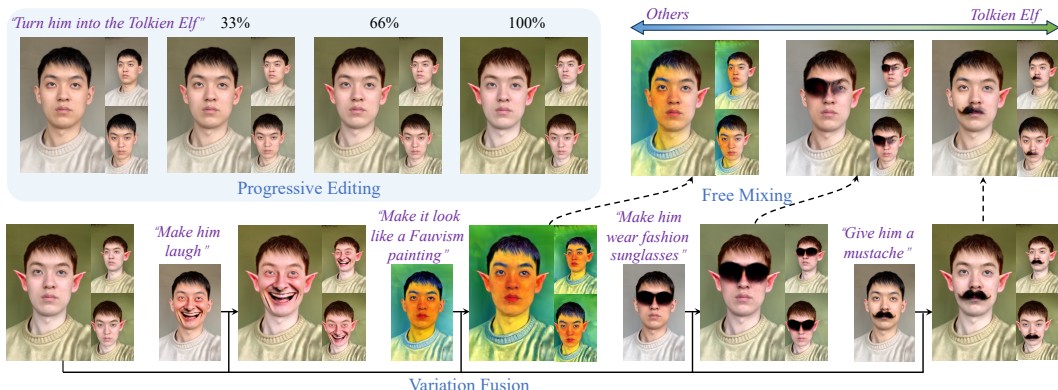

Figure 6: Schematic of the flexible editing process. The operation demonstrated in Free Mixing involves blending two set variations along the x-axis with different weights. In practical uses, the range and intensity of variations can be adjusted to personal needs to control the results.

as reconstructed data for DGE. We observe that although the carefully designed attention injection strategy significantly improves consistency across different views, discrepancies in the size of the replacement target (red ball) under various views still persist. These discrepancies led to distortions in the final reconstructed ball.

Table 2 displays a quantitative comparison between VF-Editor and baselines. Due to the extensive time required for editing by baselines, we randomly select 100 3D-instruction pairs for testing in GObj and RObj, respectively. It is evident that on RObj and GObj, DGE achieves significantly higher $C_{sim}$ and $C_{con}$ than I-gs2gs, yet its $IS$ is considerably lower. We speculate that this is due to the consistency constraints across different views, which reduce the diversity of the results. In contrast, VF-Editor, by accommodating rather than restricting diversity, significantly promotes diversity while ensuring editing quality. Furthermore, attaining the highest $IAA$ indicates that our method's editing results are more aligned with human preferences. Additionally, we find that the convergence process of $\mathcal{P}_\theta$ is hardly affected when facing multi-domain data, demonstrating its universality. In subsequent experiments, we train $\mathcal{P}_\theta$ exclusively on multi-domain data.

## 4.3 ABLATION STUDY

We design a set of iterative parallel decoding functions to decode the variation, and now we conduct ablation experiments to verify their effectiveness. Firstly, we modify iterative decoding to direct decoding, with results shown in Table 3 and Figure 4. It is evident that direct decoding fails to effectively achieve the expected goals when dealing with instructions that require

Table 3: Quantitative results of the ablation experiments on the iterative parallel decoding functions.

| Method | $IS\uparrow$ | $C_{sim}\uparrow$ | $C_{con}\uparrow$ | $IAA\uparrow$ |
|---|---|---|---|---|
| Direct Decoding | **4.71** | 0.254 | 0.801 | 5.21 |
| Triplane | 4.57 | 0.246 | 0.782 | 5.09 |
| VF-Editor-M | 4.66 | **0.259** | **0.803** | **5.22** |

Figure 7: Experimental results on the unseen data. It can be observed that the $\mathcal{P}_\theta$ trained within VF-Editor exhibits a certain degree of generalization capability, demonstrating its potential.

displacement of 3D Gaussians (although the quantitative metrics appear relatively unchanged). In contrast, the results for editing instructions that only modify the appearance of 3D Gaussians are almost unaffected. We hypothesize this is primarily due to the intercoupling of various unstructured attributes within the 3D Gaussians. If all attributes are changed simultaneously, the model tends to alter the appearance of the 3D Gaussians to meet demands rather than moving them.

Secondly, we replace the parallel decoding function with the triplane for further experimentation, and the corresponding results are presented in Table 3 and Figure 5. To visually observe the differences, we specifically select a triplet from the training set and use the stored noise as input to generate outputs corresponding to the reference edited image. It is evident that using the triplane to represent the variation field results in less distinct boundaries between different regions in the output, creating an overall blurred state. We hypothesize that this is primarily because 3D Gaussians that are spatially proximate tend to extract highly similar features from the triplanes, which are then decoded into similar variations. This issue becomes increasingly severe as the number of 3D Gaussians in the scene increases. Our parallel decoding function does not impose any prior constraints between adjacent 3D Gaussians, allowing for the learning of more refined variations.

## 4.4 FLEXIBILITY OF THE EDITING PROCESS

Compared to traditional methods, one advantage of VF-Editor is its ability to achieve flexible editing effects. Thanks to the interpretability of the variations, we can manipulate the variations in various ways to achieve the desired effects, such as merging variations, controlling the intensity of variation, and selecting local variations. As shown in Figure 6, we perform various manipulations on the variations generated by $\mathcal{P}_\theta$, resulting in diverse results: 1) Different editing strengths can be achieved by scaling the variations; 2) variations generated from different instructions can be combined to produce new editing results; 3) users can freely give different editing effects to different areas.

## 4.5 GENERALIZATION CAPABILITIES

To verify that $\mathcal{P}_\theta$ possesses generalization capabilities rather than merely memorizing the distilled knowledge of 2D editing, we conduct experiments on the unseen sample and instruction. We collect a new test set comprising 50 reconstructed objects, 50 generated objects and 10 generated scenes. As shown in Figure 7, we test the generalization of $\mathcal{P}_\theta$ obtained through distillation using $\mathcal{L}_{din}$ and $\mathcal{L}_{sds}$, respectively.

Table 4: Quantitative results on the training and test data. There is a slight decline in the quality of the editing results on the test set, but it still remains at a good level.

| Dataset | $IS\uparrow$ | $C_{sim}\uparrow$ | $C_{con}\uparrow$ | $IAA\uparrow$ |
|---|---|---|---|---|
| Training Set | **4.69** | **0.268** | **0.795** | **5.24** |
| Test Set | 4.56 | 0.241 | 0.790 | 5.16 |

The corresponding quantitative results are presented in Table 4. It can be seen that $\mathcal{P}_\theta$ demonstrates commendable generalization capabilities. Furthermore, we observe that $\mathcal{P}_\theta$ is capable of predicting reasonable editing results when the test instruction is semantically similar to one present in the training set.

## 4.6 DISCUSSION AND PROSPECTS

We observe that using $\mathcal{L}_{sds}$ alone causes the model to collapse to a single solution per instruction, while naively combining it with $\mathcal{L}_{din}$ leads to divergence. We attribute this to the nature of SDS,

which provides implicit verification rather than explicit supervision. Nonetheless, $\mathcal{L}_{sds}$ allows $\mathcal{P}_\theta$ to learn a robust baseline with good generalization, without requiring offline triplet collection. Effectively integrating $\mathcal{L}_{din}$ and $\mathcal{L}_{sds}$ may further enhance VF-Editor's capabilities. Additionally, we collect 3,348 3D-instruction pairs to train $\mathcal{P}_\theta$. Although the model generalizes well to unseen in-domain data, it does not yet support out-of-domain editing. In future work, we aim to expand the knowledge coverage of $\mathcal{P}_\theta$ efficiently. Lastly, while VF-Editor demonstrates effectiveness in object addition, relocating existing primitives may occasionally have a minor impact on surrounding regions. Introducing a dedicated primitive generation branch may further improve performance.

## 5    CONCLUSION

We present VF-Editor, a novel framework for flexible editing of 3D Gaussians by predicting spatial and appearance variations. It distills multi-source editing knowledge into a unified variation predictor, enabling precise editing across multiple scenes and instructions. To support efficient variation prediction, we introduce a variation field generation module and a set of iterative parallel decoding functions. VF-Editor offers a hopeful direction for real-time 3D editing in open-vocabulary settings.

## 6    ACKNOWLEDGEMENTS

This work was supported by the National Natural Science Foundation of China under Grant 42394060 and 42394064.

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

APPENDIX

## A  OVERVIEW OF THE APPENDIX

Here, we supplement some detailed information that is not explained in the main text but may be of interest to the reader, including:

- App. B: Some preliminary knowledge;
- App. C: Rules for drawing 3D variations;
- App. D: More training details;
- App. E: All instructions used in our training;
- App. F: More details about our custom dataset;
- App. G: Additional experimental results;
- App. H: Use of LLMs;
- App. I: Boundary conditions of generalization;
- App. J: In-depth analysis of some structural designs;
- App. K: Fine-tuning with fewer triples;
- App. L: Further evaluation;
- App. M: Runtime comparison.

In addition, we provide the Demo.mp4 in the supplementary materials and simultaneously upload source code to the anonymous repository: `https://anonymous.4open.science/r/VF-Editor`

**Ethics Statement** This research uses only publicly available datasets that contain no personally identifiable or sensitive information. No human or animal subjects are involved. All experiments are conducted in accordance with institutional and conference ethical guidelines.

**Reproducibility Statement** We provide detailed descriptions of the model architecture and hyperparameter settings in Section 4.1 and App. D. Our code will be released to facilitate reproducibility.

## B  PRELIMINARY KNOWLEDGE

### B.1  3D GAUSSIAN SPLATTING

3D Gaussian Splatting has revolutionized the field of 3D representation, employing anisotropic 3D Gaussians for efficient and intricate radiance fields modeling Kerbl et al. (2023). It explicitly represents the radiance field as a mixture of Gaussians $\mathcal{P} = \{(\mu_i, \alpha_i, s_i, c_i, r_i)\}_i^N$, where $\mu_i \in \mathbb{R}^3$ is the mean, $\alpha_i \in \mathbb{R}$ is the opacity, $s_i \in \mathbb{R}^3$ is the scale matrix, $c_i \in \mathbb{R}^3$ is the view-dependent RGB color computed from Spherical Harmonic coefficients, $r_i \in \mathbb{R}^{3 \times 3}$ is the rotation matrix. With $x$ denoting the queried point, and $\Sigma_i$ denoting the Gaussian's covariance matrix calculated as $\Sigma_i = r_i s_i s_i^T r_i^T$, the Gaussian function is defined as:

$$G_i(x) = e^{-\frac{1}{2}(x-\mu_i)^T \Sigma_i^{-1}(x-\mu_i)} \tag{7}$$

In the rasterization process, the 3D Gaussians are splatted to screen-space 2D Gaussians following EWA Splatting Zwicker et al. (2001). Denoting $u$ as the given screen point, $\Sigma_i'$ as the covariance matrix of the 2D Gaussian, the screen-space Gaussian function is $g_i(u) = e^{-\frac{1}{2}(u-\mu_i)^T \Sigma_i'^{-1}(u-\mu_i)}$. The splatted Gaussians are blended following the volumetric rendering model formulated as:

$$C = \sum_{i \in N} c_i o_i \prod_{j=1}^{i-1}(1 - o_i) \tag{8}$$

where $o_i = \alpha_i g_i(u)$ for the given screen point $u$. The process is implemented with a tile-based CUDA rasterizer, which allows real-time differentiable rendering of 3D Gaussian Splatting.

## B.2 DDIM INFERENCE

Denoising Diffusion Implicit Models (DDIM) Song et al. (2021) reformulate the stochastic reverse process of DDPM Ho et al. (2020) into a deterministic non-Markovian mapping, enabling high-quality samples with far fewer steps. Let

$$\bar{\alpha}_t = \prod_{i=1}^{t}(1 - \beta_i), \quad \bar{\alpha}_0 = 1, \tag{9}$$

and let $\varepsilon_\phi(x_t, t)$ denote the model's predicted noise. The DDIM update from $x_t$ to $x_{t-1}$ is then given by

$$x_{t-1} = \sqrt{\bar{\alpha}_{t-1}} \, \frac{x_t - \sqrt{1 - \bar{\alpha}_t}\, \varepsilon_\phi(x_t, t)}{\sqrt{\bar{\alpha}_t}} \; + \; \sqrt{1 - \bar{\alpha}_{t-1}}\, \varepsilon_\phi(x_t, t). \tag{10}$$

Crucially, by setting the variance of each transition to zero, DDIM produces a fully deterministic trajectory. During inference, one can further "skip" intermediate timesteps—selecting only $S \ll T$ of the original $T$ steps—which typically yields perceptually comparable images with an order-of-magnitude reduction in sampling cost. The determinism of DDIM not only makes sample generation reproducible but also facilitates smooth interpolation between latent representations.

## B.3 DIFFUSION INVERSION

In the original DDPM framework Ho et al. (2020), the forward noising process is defined by

$$q(x_t \mid x_{t-1}) = \mathcal{N}\big(x_t; \sqrt{1 - \beta_t}\, x_{t-1}, \beta_t \mathbf{I}\big), \tag{11}$$

which admits the closed-form marginal

$$q(x_t \mid x_0) = \mathcal{N}\Big(x_t; \sqrt{\bar{\alpha}_t}\, x_0, (1 - \bar{\alpha}_t)\mathbf{I}\Big), \quad \bar{\alpha}_t = \prod_{i=1}^{t}(1 - \beta_i). \tag{12}$$

To invert a given image $x_0$ back into its noise latent $x_T$, one simply "runs" this forward process: for $t = 1, \ldots, T$,

$$x_t = \sqrt{\bar{\alpha}_t}\, x_0 + \sqrt{1 - \bar{\alpha}_t}\, \varepsilon_t, \quad \varepsilon_t \sim \mathcal{N}(0, \mathbf{I}). \tag{13}$$

We obtain the ground truth latent at different noise levels $x_{[0:T]}$. With these noised latents, we can correct the trajectory of the backward process by aligning the predicted previous sample $\tilde{x}_{t-1}$ with the ground truth previous sample $x_{t-1}$ to obtain the variational noise in each DDIM step:

$$\tilde{x}_{t-1} = \frac{\sqrt{\bar{\alpha}_{t-1}}}{\sqrt{\bar{\alpha}_t}}\big(x_t - \sqrt{1 - \bar{\alpha}_t}\, \varepsilon_\phi(x_t, y, t)\big) + \sqrt{1 - \bar{\alpha}_{t-1} - \sigma_t^2} \cdot \varepsilon_\phi(x_t, y, t).$$

$$\tilde{\varepsilon}_t = \frac{x_{t-1} - \tilde{x}_{t-1}}{\sigma_t}. \tag{14}$$

By setting $\sigma_t = \sqrt{(1 - \bar{\alpha}_{t-1})/(1 - \bar{\alpha}_t)}\sqrt{1 - \bar{\alpha}_t/\bar{\alpha}_{t-1}}$ for all $t$ to make the sampling process align with the original DDPM, we can obtain edited latents $x'_{[0:T]}$ with the following equations

$$x'_T = x_T$$

$$x'_{t-1} = \frac{\sqrt{\bar{\alpha}_{t-1}}}{\sqrt{\bar{\alpha}_t}}\big(x'_t - \sqrt{1 - \bar{\alpha}_t}\, \varepsilon_\phi(x'_t, y', t)\big) + \sqrt{1 - \bar{\alpha}_{t-1} - \sigma_t^2} \cdot \varepsilon_\phi(x'_t, y', t) + \sigma_t \cdot \tilde{\varepsilon}_t \tag{15}$$

under new conditioning $y'$ (*e.g.* text, masks, or style codes), enabling precise, content-preserving edits of the input image.

## B.4 SCORE DISTILLATION SAMPLING

Score Distillation Sampling (SDS) Poole et al. (2022) is a technique that turns a pretrained diffusion model into a powerful "teacher" for training a separate, fully differentiable network—often used to represent 3D scenes or other complex modalities. Instead of learning from raw data, the student network learns to match the denoising behavior of the diffusion model: at each noise level, it adjusts its parameters so that its own predictions of what a clean sample should look like align with those

of the frozen diffusion teacher (as shown in Equation 6). By doing this across all noise scales, SDS effectively transfers the rich generative knowledge embedded in the diffusion model—without ever retraining it—enabling the student network to produce samples that the teacher considers highly plausible. This approach unlocks efficient, high-fidelity synthesis in new domains (*e.g.*, text-to-3D) by distilling the diffusion model's learned prior into a specialized downstream network.

## C  3D VARIATION VISUALIZATION

For the purpose of observation, we present the 2D projections of the 3D variations in the main text. The projection rules for different attributes are as follows:

**1) Spatial Position** $\mu$: We project the original and edited positions of each 3D Gaussian onto a 2D plane and connect these two points with a line segment to represent the change in spatial position. The opacity of the line segment indicates the magnitude of the displacement, with higher opacity corresponding to a larger displacement. The color of the line segment represents the direction of the displacement. We uniformly set three standard direction vectors on the plane, each corresponding to pure red, pure green, and pure blue, and then interpolate to obtain the color corresponding to a specific direction.

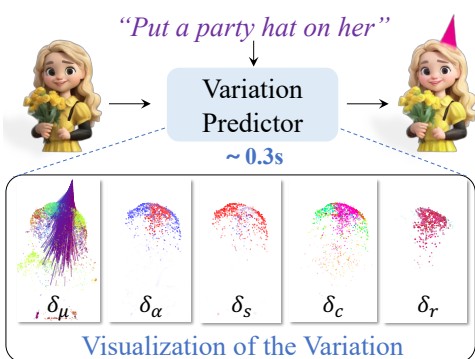

Figure 8: 2D visualization of the 3D variation.

**2) Opacity** $\alpha$: We project the original position of each 3D Gaussian onto a 2D plane and draw a circle at the projection point to represent the change in opacity. If the opacity increases, the circle is colored red; if the opacity decreases, the circle is colored blue; if the change in opacity is small, the circle appears close to white. As shown in Figure 8, to form *paty_hat*, a large number of 3D Gaussians around the head of the character have increased in opacity.

**3) Scaling Coefficient** $s$: Similar to opacity, we use red to represent an increase in the scaling coefficient and blue to represent a decrease. For convenience in presentation, we take the average of the scaling coefficients along the x, y, and z axes before proceeding with subsequent calculations.

**4) Color** $c$: We present the variation of each 3D Gaussian's color rather than its resultant quantity. Specifically, we project the original position of each 3D Gaussian onto a 2D plane and then draw a circle at the projected point to represent the color variation. The opacity of the circle is related to the magnitude of the color change; the less the color variation, the more transparent the circle. Additionally, we scale and translate the RGB variation to the range of 0 to 1 to represent the color of the circle. As shown in Figure 8, the colors of a large number of 3D Gaussians at the head transition to red, forming a *paty_hat*.

**5) Rotation Quaternions** $r$: We find that simple projection strategies fail to intuitively represent the variation process of rotation quaternions in a 2D plane. Therefore, we do not display this process in the main text. As shown in Figure 8, we map the first component of the change of quaternion to opacity and the remaining three components to color.

## D  MORE TRAINING DETAILS

Here, we provide additional details regarding the training of VF-Editor-M. VF-Editor-S, in comparison, only differs by a reduced number of training epochs without any other modifications. For $\mathcal{L}_{\text{din}}$, the model is trained on the collected triplets for 600 epochs. The initial learning rate is set to 1e-4 and halves every 100 epochs. The AdamW Zhou et al. (2024) optimizer is used, with the weight decay set to 5e-3. For $\mathcal{L}_{\text{sds}}$, the model is trained on all 3D-instruction pairs for 500 epochs. The initial learning rate is set to 1e-4 and halves every 100 epochs. The AdamW optimizer is also used, with the weight decay set to 5e-3. For IP2P, the *guidance_scale* and *condition_scale* are set to 7.5 and 1.5, respectively. The noise schedule time step $t$ decreases linearly as the number of epochs increases.

Table 5: 3D-instruction pairs used in the collection of triplets.

| Type | Instruction |
|---|---|
| RObj | make it look like it's covered in moss |
| | make its color look like rainbow |
| | make its color look like gold |
| | make it look wooden |
| GObj | Turn him into a clown |
| | Put a party hat on him |
| | Turn him into the Tolkien Elf |
| | Turn his hair orange |
| | Turn his clothes blue |
| | Turn his pants green |
| | Put a party hat on her |
| | Turn her into the Tolkien Elf |
| | Turn her into a clown |
| Scene | Make it look like a Van Gogh painting |
| | Make him a bronze statue |
| | Make it look like a Fauvism painting |
| | Give him red hair |
| | Make him a marble statue |
| | Replace the sunflower with a red ball |
| | Make it colorful |

We leverage an aesthetic scorer Yi et al. (2023) to filter out low-quality edited results during the triplet collection process. Specifically, for each instruction targeting a particular subset, after gathering a batch of triplets, we use Yi et al. (2023) to evaluate all edited images and retain only those triplets whose edited images rank within the top 50% of scores.

For the Variation Field Generation Module $\mathcal{M}$, we set the number of attention heads to 16, with a head dimension of 64. For the Iterative Parallel Decoding Function $\mathcal{F}$, we use a single attention head with a head dimension of 64. The output dimensions of $\mathcal{F}_1$ and $\mathcal{F}_2$ are set to 3 and 11, respectively. In the collected triplets, the noise has a shape of $1*4*64*64$, and the image has a shape of $1*3*512*512$. To strike a balance between efficiency and visual quality, we set the number of sampling steps in the diffusion model to 50. During data collection, we employed multiple diffusion models, including IP2P Brooks et al. (2023), CtrlColor Liang et al. (2024), and Stable Diffusion 2.1 (Inversion) Rombach et al. (2022a). When computing $\mathcal{L}_{sds}$, we set *guidance_scale* and *condition_scale* to 6.5 and 3.5, respectively, and linearly decay the time step $t$ from 800 to 100 over the course of training.

# E  ALL INSTRUCTIONS

We collect 32,566 triplets through DDIM inference and Diffusion inversion, employing 3D-Instruction pairs as shown in Table 5 during the collection process. Additionally, we design more instructions while using $\mathcal{L}_{sds}$ to train $\mathcal{P}_\theta$, as illustrated in Table 6.

Table 6: 3D-instruction pairs used during training with $\mathcal{L}_{\text{sds}}$.

| Type | Instruction |
|---|---|
| GObj | Make him wear fashion sunglasses |
| | Turn him into the Tolkien Elf |
| | Make her wear fashion sunglasses |
| | Turn her into the Tolkien Elf |
| Scene | Make him wear fashion sunglasses |
| | Give him a mustache |
| | Turn him into the Tolkien Elf |
| | Make him laugh |
| | What would he look like as a bearded man |
| | Give him a cowboy hat |

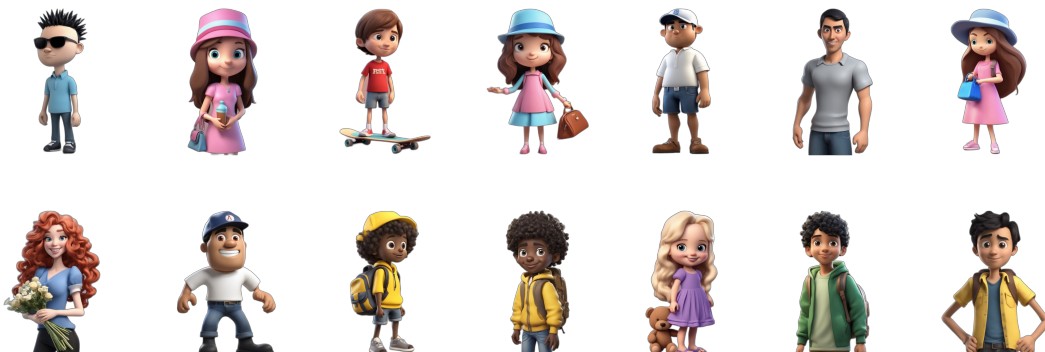

Figure 9: Some images generated by SD3.

## F    CUSTOM DATASET

To train $\mathcal{P}_\theta$, we collect a set of 3D data. The number of 3D Gaussians contained in each custom data ranges approximately from 5,000 to 50,000. Here, we describe the data from the GObj subset and part of the data from the Scene subset.

**Generated Objects (GObj):** First, we utilize GPT-4 Achiam et al. (2023) to generate instructions describing cartoon characters with different appearances. These instructions are then input into SD3 Esser et al. (2024) to generate the corresponding images. Finally, we use the image-to-3D generation model V3D Chen et al. (2024c) to convert these images into 3DGS. A portion of the images generated by SD3 is shown in Figure 9. We will release all images and 3DGS data publicly.

**Reconstructed Scenes (Scene):** We construct three sets of custom scene data to further evaluate the generalization capability of VF-Editor. These include: 1) `doll_grayscale`: We initially capture 203 images of a dinosaur doll and crop their dimensions to 512x512. Then, we use COLMAP Schonberger & Frahm (2016) to compute the pose of each image and convert all images to grayscale. We employ Mini-splatting Fang & Wang (2024) for the reconstruction of these images, with the $sh$ degree set to 0 and the $sampling\,factor$ set to 0.1. 2) `sunflower_grayscale`: This scene contains 197 training images, with the remaining processing steps being the same as those in `doll_grayscale`. 3) `sunflower`: This scene is the color version of `sunflower_grayscale`. Some of the training images from these three scenes are shown in Figure 10.

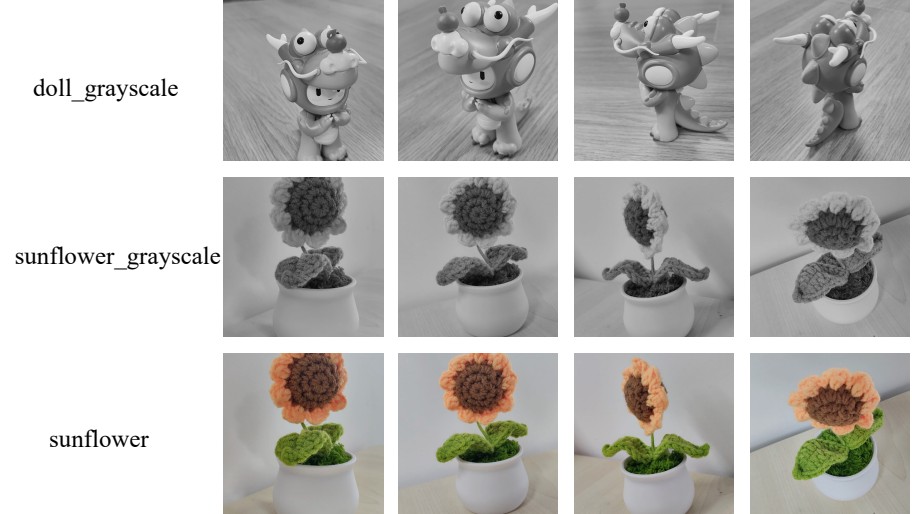

Figure 10: Some images from the custom scene data.

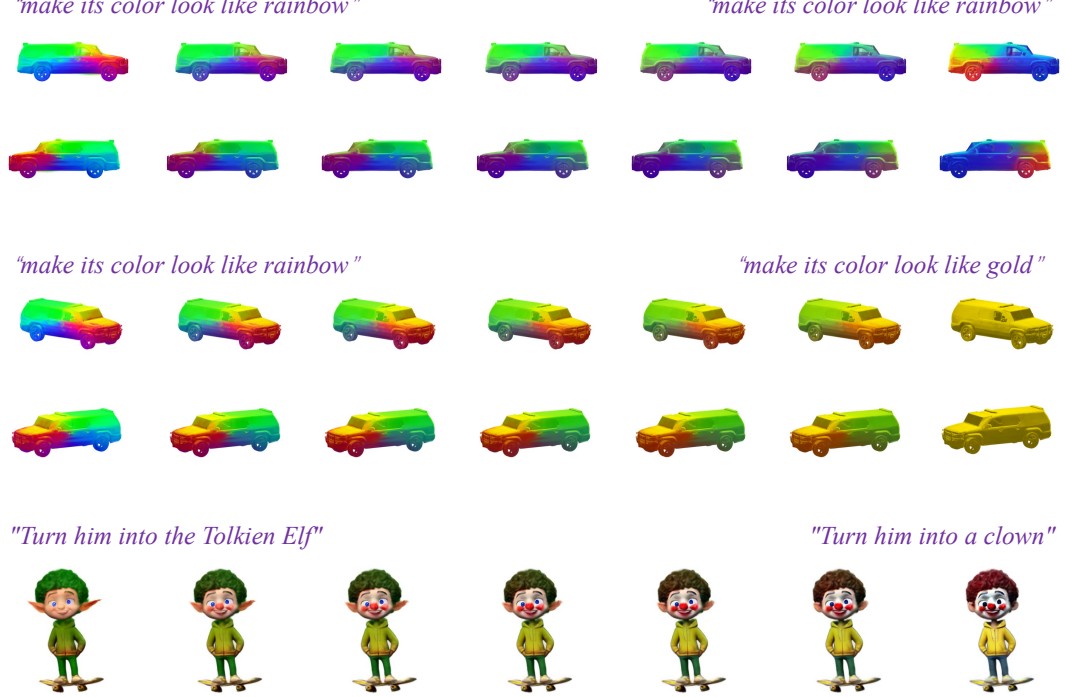

Figure 11: Interpolation of variations.

# G ADDITIONAL EXPERIMENTAL RESULTS

## G.1 INTERPOLATION OF VARIATIONS

In the main text, we provide the editing results obtained after blending the variations. Here, we supplement with more detailed interpolation results of the variations to validate the smoothness of the generated variations. The experimental results are shown in Figure 11, where it can be observed that interpolating between the two variations with different weights produces natural intermediate results. In *Demo.mp4*, we provide additional editing results.

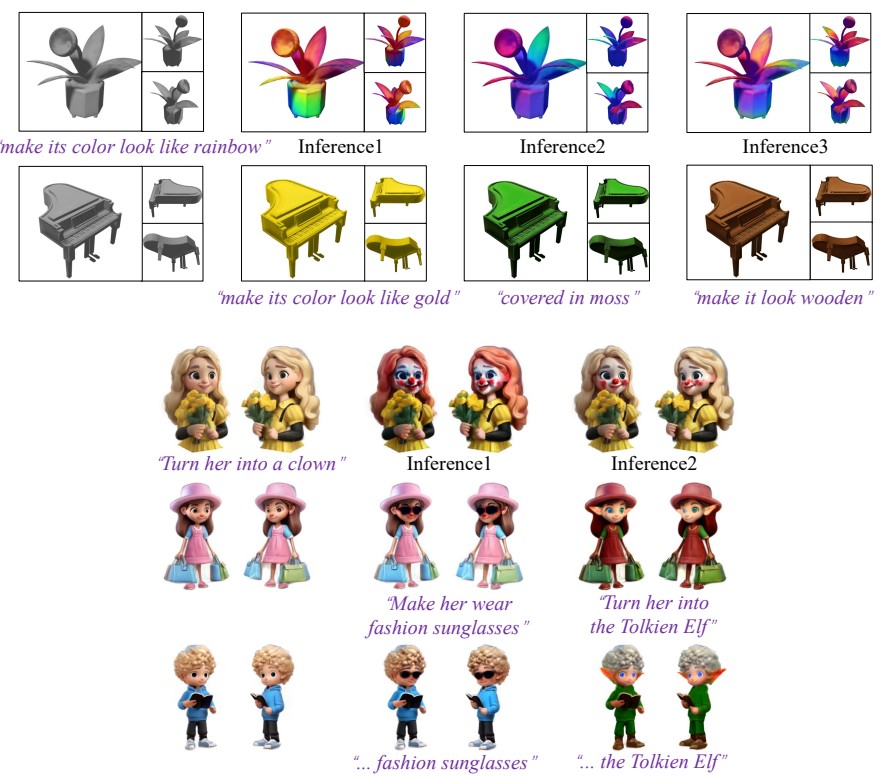

Figure 12: Results of multiple rounds of editing on the same sample.

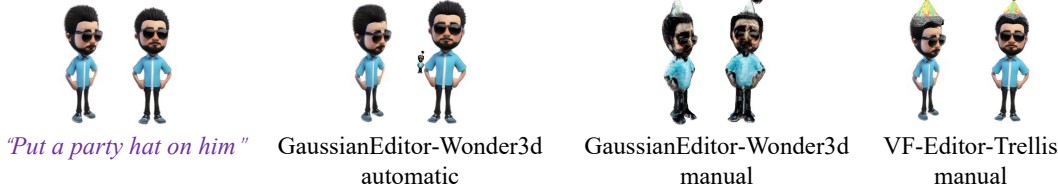

Figure 13: Comparison with GaussianEditor.

## G.2    Multiple Inferences

In Figure 12, we present the results of multiple inference runs on a specific 3D model under identical and varying instructions to evaluate the diversity and stability of our method's editing outputs.

## G.3    Comparison with GaussianEditor

GaussianEditor Chen et al. (2024b) proposes the optional integration of an external 3D generator Long et al. (2024) to assist with object insertion tasks. However, considering that none of the other baselines utilize such auxiliary generators, we adopt the pure GaussianEditor algorithm for fair comparison in Figure 3 and Table 2. Here, we additionally present editing results that incorporate the auxiliary generator, as illustrated in Figure 13. It can be observed that relying solely on the automatic pipeline of GaussianEditor struggles to produce semantically reasonable object insertions. When the object location is manually specified, the default holistic optimization often compromises the appearance of the original scene. In contrast, we also present results using the external Trellis Xiang et al. (2025) module, where the inserted object is manually placed in a semantically coherent location, serving as a visual reference for comparison.

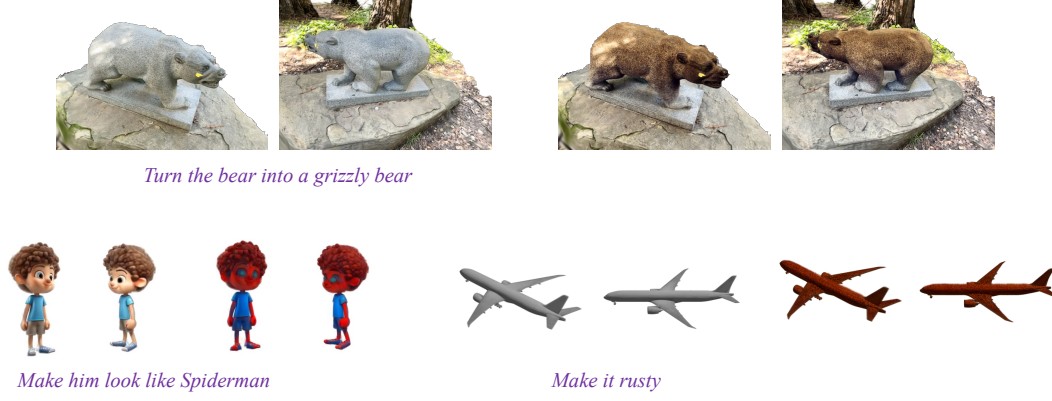

*Turn the bear into a grizzly bear*

*Make him look like Spiderman*        *Make it rusty*

Figure 14: Fine-tuning results on unseen instructions.

Table 7: The results of different grouping strategies.

|  | $IS\uparrow$ | $C_{sim}\uparrow$ | $C_{con}\uparrow$ | $IAA\uparrow$ |
|---|---|---|---|---|
| Random sampling | 4.24 | 0.227 | 0.881 | 5.27 |
| Farthest point sampling | 4.05 | 0.196 | 0.973 | 5.20 |
| Spatial-color k-means | 4.21 | 0.227 | 0.879 | 5.22 |

## G.4 FINE-TUNING RESULTS

Figure 14 presents the results of model fine-tuning on three novel instructions not encountered during pretraining. For each out-of-domain instruction, fine-tuning for 10–20 hours enables the model to acquire the desired new editing capability without compromising its original performance. After fine-tuning, similar editing operations can be performed in under 0.3 seconds.

## G.5 MORE ABLATION EXPERIMENTS

To further investigate the impact of the tokenizer on model performance, we conduct two ablation studies on the GObj subset.

**(1) Changing the grouping strategy in the tokenizer.**

We implemented two new grouping methods: (i) selecting anchor points using farthest point sampling (FPS), a technique commonly used in point-cloud models, followed by clustering; and (ii) spatial–color k-means clustering, where both the color coefficients and the positions of the Gaussians are equally weighted (0.5). The experimental results are shown in table 7. We observe that FPS leads to the weakest performance, while the remaining two methods yield comparable results.

We hypothesize that this is mainly because in 3DGS scenes, the Gaussian primitives are typically denser in central regions with complex structures and sparser near scene boundaries. Due to its distance-aware mechanism, FPS tends to select anchor points along the scene's global contour. As a result, sparse boundary primitives are frequently chosen as anchors, leading to a relatively fixed and uneven distribution of anchor points. In contrast, uniform (random) sampling does not consider spatial distances and therefore selects anchors more frequently from the dense central regions, resulting in a more diverse distribution of anchor points. In addition, although the performance of random sampling and spatial–color k-means clustering is similar, random sampling followed by clustering requires substantially less computation.

**(2) Further analysis of group size.**

We then fix the anchor sampling strategy to random sampling and vary the group size (adjusting the number of groups proportionally) to examine its influence on model performance. As shown in table 8, within a reasonable range, different group sizes have only a minor impact on performance.

Table 8: The results of the ablation study on group size.

| Group size | $IS\uparrow$ | $C_{sim}\uparrow$ | $C_{con}\uparrow$ | $IAA\uparrow$ |
|---|---|---|---|---|
| 64 | 4.24 | 0.228 | 0.879 | 5.27 |
| 128 | 4.24 | 0.227 | 0.881 | 5.27 |
| 192 | 4.22 | 0.223 | 0.876 | 5.24 |

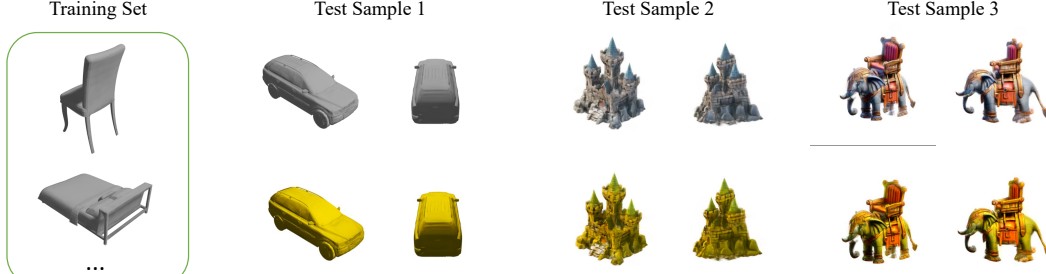

Instruction: *"make its color look like gold"*

Figure 15: Experimental results of fixing the instruction while changing the edited 3D model.

## H   USE OF LLMS

In our manuscript, we employed the LLM to perform grammatical checks on the written content. Furthermore, we utilized GPT-4 to generate the image synthesis prompts required for our study, as detailed in Section F.

## I   BOUNDARY CONDITIONS OF GENERALIZATION

Here, we investigate the boundary conditions of generalization. We conduct two sets of experiments: (1) fixing the instruction while changing the edited 3D model, and (2) fixing the 3D model while varying the semantics of the instruction. As shown in Figure 15 and 16, we observe the following: (1) For general-purpose instructions, the model is able to produce reasonable editing results even on out-of-domain data. (2) As the test instruction gradually deviates semantically from the original one (e.g., "*make its color look like a rainbow*"), the model's ability to follow the instruction also degrades. In Figure 16, when the test instruction remains semantically related to the training instructions—such as "*apply a vivid spectrum of colors*"—the editing outputs remain reasonable and exhibit a distribution distinct from the outputs obtained using the original training instruction. However, when the semantics of the test instruction differ substantially from those seen during training—for instance, "*make the lighting dramatic and moody*"—the model's behavior begins to diverge from the instruction. We suspect this occurs because the model lacks the ability to generalize to entirely unseen semantic concepts that were never learned during training.

## J   IN-DEPTH ANALYSIS OF SOME STRUCTURAL DESIGNS

Here, we provide additional in-depth analyses related to the structural design.

**(1) Why use transformer blocks *with* self-attention in the Variation Field Generation Module ($\mathcal{M}$), but *without* self-attention in the Iterative Parallel Decoding Function ($\mathcal{F}$)?**

Our Variation Predictor can be viewed as a *structured factorization* of the 3D editing operator:

- $\mathcal{M}$ is responsible for extracting global correlations from the input scene and generating a *global variation field*.

- $\mathcal{F}$ then performs lightweight *per-primitive* update decoding for each Gaussian primitive.

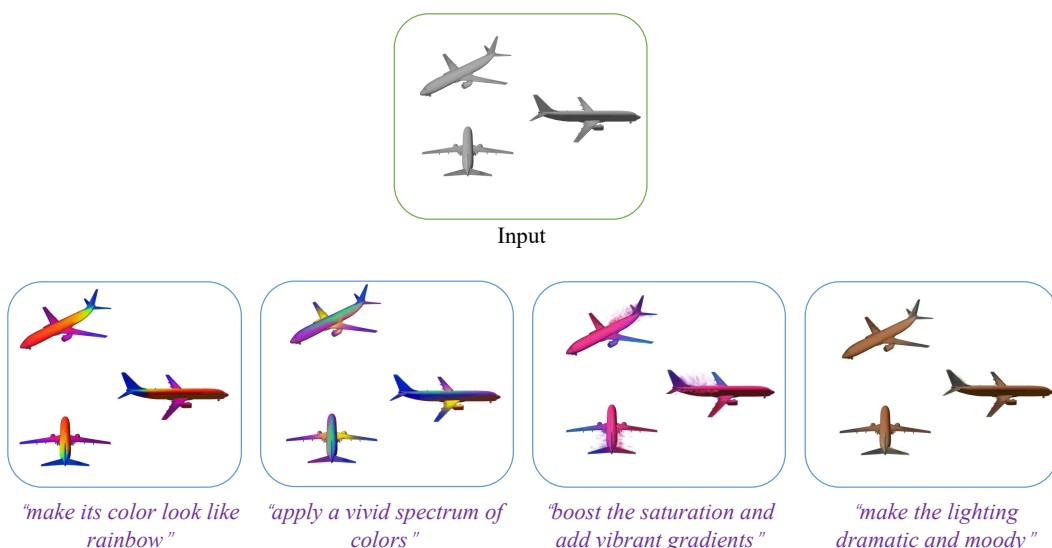


Input


*"make its color look like rainbow"*    *"apply a vivid spectrum of colors"*    *"boost the saturation and add vibrant gradients"*    *"make the lighting dramatic and moody"*

Figure 16: Experimental results of fixing the 3D model while varying the semantics of the instruction.

This factorization is analogous to the common encoder–latent–decoder structure in representation learning: the variation field $f_\Delta$ serves as a global, low-dimensional, cross-primitive *latent basis*, while $\mathcal{F}_1/\mathcal{F}_2$ are only responsible for conditional decoding (conditional readout).

Under this design:

- **Why $\mathcal{M}$ needs self-attention.** The variation field must aggregate structural information from the *entire* scene. Given the presence of a tokenizer, self-attention is the most effective global information aggregation mechanism, as it can capture cross-primitive correlations with a controllable computational budget.

- **Why $\mathcal{F}$ intentionally avoids self-attention.** Once a global variation field is available, introducing self-attention inside $\mathcal{F}$ would force all Gaussian primitives to interact pairwise. This would increase the complexity from $O(N)$ to $O(N^2)$, which becomes prohibitively expensive for large-scale scenes. At the same time, we would like each primitive's update to remain *conditionally independent* given the global field so that decoding can be fully parallelized. Therefore, $\mathcal{F}$ uses only cross-attention, conditioning on the same shared variation field. This keeps the decoding complexity linear in $N$, which is crucial for large scenes.

**(2) Why do we separate $\mathcal{F}_1$ (mean) from $\mathcal{F}_2$ (scale/opacity/color/rotation)?**

This design is motivated by the analytical structure of the 3DGS render (Eq. 7–8 in App. B). The rendered color $C = f(\mu, s, \alpha, c, r)$ admits the following Jacobian under a first-order Taylor expansion with respect to the parameters:

$$J_\theta = \begin{bmatrix} J_{\mu\mu} & J_{\mu A} \\ J_{A\mu} & J_{AA} \end{bmatrix}, \quad A = (s, \alpha, c, r).$$

The non-zero off-diagonal blocks indicate that the geometric position $\mu$ and the appearance attributes $A$ are *intercoupled*:

- Changing $\mu$ alters the distribution of primitives in screen space, which in turn affects the effective contribution of their opacity/scale/color.

- Changing appearance attributes likewise affects how gradients propagate along geometric directions.

Existing 3DGS reconstruction works Charatan et al. (2024); Lan et al. (2025) have also adopted similar strategies to avoid letting the appearance parameters (scale, color, opacity) "absorb the error."

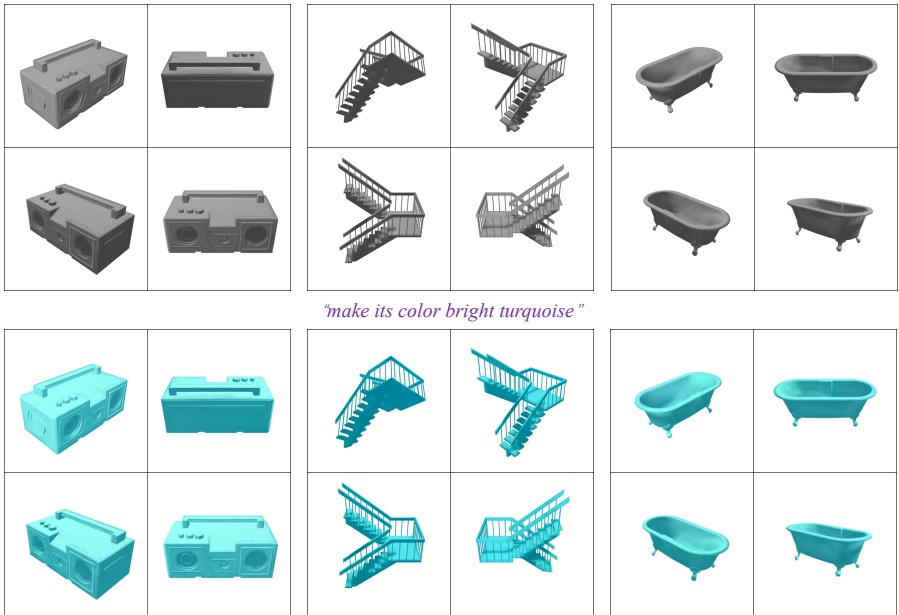

Figure 17: Fine-tuning with fewer triples.

Against this background, our $\mathcal{F}_1/\mathcal{F}_2$ design can be interpreted as an *approximate block-coordinate update / block-diagonalization* of the Jacobian:

- $\mathcal{F}_1$ first predicts the geometric position $\mu$ preventing geometric gradients from being overwhelmed or distorted by appearance parameters.
- $\mathcal{F}_2$ then updates the appearance attributes after the geometry has been stabilized.

This structure alleviates the optimization instability caused by the strong coupling between geometry and appearance, enabling the system to handle both *geometric edits* and *appearance edits* more reliably.

As shown in Figure 4 and Table 3:

- Directly decoding all attributes jointly is prone to failure under displacement-type edits.
- Our iterative decoding remains stable across almost all types of edits.

## K    FINE-TUNING WITH FEWER TRIPLES

It is foreseeable that once the model has learned a particular category of edit, acquiring unseen concepts within the same category should become substantially easier. To verify this intuition, we conduct a simple experiment. We introduce a new instruction, "*make its color bright turquoise*", containing the unseen concept "bright turquoise". Following the procedure described in Section 3.2.2, we collect only 25 triplets for this new instruction and fine-tune the model. As shown in Figure 17, the model successfully learn the new concept with ease. This demonstrates that for edits within the same category, even novel concepts can be learned with very few training examples. For instance, once the model has learned "*make its color look like gold*," learning "*make its color bright turquoise*" becomes much easier.

## L    FURTHER EVALUATION

To further validate the effectiveness of our method, we evaluate the editing results from multiple perspectives.

Table 9: Consistency scores across views computed using GPT-5.

| Method | $GPT-score_{edit}$ ↑ |
|---|---|
| I-gs2gs | 6.3 |
| GaussianEditor | 6.7 |
| DGE | 7.5 |
| VF-Editor-M | 8.3 |

Table 10: The results of the user study.

| Method | $Aesthetic\ Quality$ ↑ | $Instruction\ Following$ ↑ | $Faithfulness$ ↑ |
|---|---|---|---|
| I-gs2gs | 4.7 | 5.1 | 6.4 |
| GaussianEditor | 5.0 | 6.0 | 5.9 |
| DGE | 6.4 | 5.9 | 7.5 |
| VF-Editor-M | 9.3 | 9.0 | 9.0 |

(1) 3D Geometry Preservation. To assess whether our edits affect the underlying 3D structure for non-geometric edits, we simplify the 3DGS representation into point clouds and compute the Chamfer Distance and F-score ($\tau = 0.01$) between the point clouds before and after applying two representative non-geometric edits ("*make its color look like rainbow*" and "*make its color look like gold*"). Across 100 trials, the average results are: Chamfer Distance = 2.4273e-05, F-score = 0.9730. These outcomes indicate that the geometry of the 3D model remains almost entirely intact.

(2) Multi-view Consistency Checks. To provide a more comprehensive assessment of view consistency, we introduce a new metric called, $GPT-score_{edit}$. We render some edited views and feed the resulting images into GPT-5, using a carefully designed prompt to score their cross-view consistency. The average scores over the 11 test cases are reported in table 9. The full prompt:

################################################################################

You are an expert 3D graphics evaluator. You will be given several rendered images of the same 3D object from different viewpoints. Your task is to assess **multi-view consistency**: how well these images appear to come from a single coherent 3D model.

**Instructions:**

1. **Consistency Definition:** Evaluate whether the object's shape, proportions, geometry, materials, textures, and global structure remain consistent across all views.
2. **Ignore Rendering Artifacts:** Do not consider lighting, shadows, background, camera angle mismatch, or rendering noise—only evaluate the intrinsic object identity and structural consistency.

3. **Scoring Rule (0–10):**

  - **0**: extremely inconsistent; views clearly depict different objects
  - **5**: moderately consistent; some mismatches exist but the main structure aligns
  - **10**: perfectly consistent; all views depict the same coherent 3D model

4. **Output Format:** Respond only with a single number from 0 to 10, representing your consistency score.

**Now evaluate the consistency of the provided images.**

################################################################################

(3) User Study. We also conduct a user study using a questionnaire-based evaluation. For each question, volunteers are shown videos rendered from the edited results of different methods, along with the corresponding editing instruction, all in randomized order. Participants rated each result in terms of Aesthetic Quality, Instruction Following, and Faithfulness to the Original 3D Model, using a scale from 0 (worst) to 10 (best). We collect scores from 11 volunteers across 5 sets of data, and the averaged results are reported in the table 10.

Table 11: The editing times of different methods.

| Time (s) | Training Set | Test Set |
|---|---|---|
| I-gs2gs | 275 | 275 |
| GaussianEditor | 463 | 463 |
| DGE | 210 | 210 |
| VF-Editor-M | 216 | **0.3** |

## M  RUNTIME COMPARISON

Here, we compare the editing times of different methods. Since all existing baselines perform optimization on a per-scene basis, we report both the amortized editing time on the training set and the editing time on the test set separately for a comprehensive comparison, as shown in table 11.

