# OpenReview forum: "Variation-aware Flexible 3D Gaussian Editing"
_ICLR.cc/2026/Conference — ICLR 2026 Poster_

### Official Review · Reviewer_giZT · 2025-10-27

**Soundness:** 3
**Presentation:** 3
**Contribution:** 3
**Rating:** 6
**Confidence:** 3

**Summary:**

The paper proposes VF-Editor, a novel method for editing 3D Gaussian Splatting (3DGS) scenes directly by predicting attribute variations. Instead of relying on indirect editing through 2D image manipulation, VF-Editor distills knowledge from 2D editors into a native 3D editor. A variation predictor, composed of a variation field generation module and parallel decoding functions, is trained to efficiently estimate attribute changes for each 3D Gaussian. The paper emphasizes the method's flexibility, efficiency, and ability to learn from diverse 2D editing strategies, leading to consistent and high-quality 3D edits.

**Strengths:**

-The core idea of directly predicting attribute variations for 3D Gaussians, guided by distilled 2D editing knowledge, is novel and addresses limitations of existing indirect editing methods.

-The design of the variation predictor, with its transformer-based components, is very fast since it is feedforward rather than relying on computationally expensive techniques.

-The parallel decoding functions promise efficient processing.

-The paper presents some qualitatively impressive results, demonstrating the ability to perform various editing operations (e.g., style transfer, object replacement) while preserving scene details.

**Weaknesses:**

-Uses a 2D diffusion prior from models like DDIM has been used in some other 3D scene editing works, such as:
[CVPR 2023] DiffusioNeRF: Regularizing Neural Radiance Fields with Denoising Diffusion Models
[NIPS 2024] In-N-Out: Lifting 2D Diffusion Prior for 3D Object Removal via Tuning-Free Latents Alignment

-A more thorough ablation of the different components in VF-Editor (e.g., contribution of the random tokenizer, importance of different loss terms) would be useful.

-While claiming "real-time editing", the paper lacks precise runtime measurements or a performance comparison against other editing methods.

**Questions:**

- Many details are missing like dimensions, size, exact steps, what noise model is used, more details on losses, etc.

- Are there specific types of edits that VF-Editor struggles with (e.g., large structural changes, view-dependent effects, highly detailed textures)? What are the failure modes?

- More elaboration on the fundamental difference with prior works that utilise 2D diffusion priors on 3D editing tasks.

---

> ### Author Response · Authors · 2025-11-22
> **Response to Weaknesses**
>
> **W1. Uses a 2D diffusion prior from models like DDIM has been used in some other 3D scene editing works.**
>
> **Response to W1:** We sincerely appreciate your careful review and for pointing out these related works. DiffusioNeRF [1] learns a diffusion prior over RGB-D patches and uses its score as an additional regularizer during per-scene NeRF optimization, while In-N-Out [2] performs 3D object removal by inpainting occluded regions in a single view with a 2D diffusion model and then propagating this prior to other views via latent alignment. Conceptually, both methods are closely related to the baseline we compare against in our paper: *they indirectly exploit a 2D diffusion prior by back-projecting the outputs of a 2D diffusion model into 3D space to modify the 3D representation.* However, *due to the black-box nature of neural networks, these approaches are inherently prone to cross-view inconsistencies, and their per-scene optimization paradigm further limits flexibility.* In contrast, **our method redefines the 3DGS editing task as a feed-forward variation prediction problem, which fundamentally avoids cross-view inconsistency and substantially improves editing flexibility.** We have now cited these two works in the Introduction to make our presentation more complete.
>
> [1] Wynn J, Turmukhambetov D. Diffusionerf: Regularizing neural radiance fields with denoising diffusion models[C]//Proceedings of the IEEE/CVF Conference on Computer Vision and Pattern Recognition. 2023: 4180-4189.
>
> [2] Hu D, Fu H, Guo J, et al. In-N-Out: Lifting 2D Diffusion Prior for 3D Object Removal via Tuning-Free Latents Alignment[J]. Advances in Neural Information Processing Systems, 2024, 37: 45737-45766.
>
> ---
>
> **W2. A more thorough ablation of the different components in VF-Editor (e.g., contribution of the random tokenizer, importance of different loss terms) would be useful.**
>
> **Response to W2:** Thank you very much for your helpful suggestion. We have added the corresponding ablation studies in Appendix G.5.
>
> **(1) Contribution of the random tokenizer.**
>
> To evaluate the contribution of the random tokenizer, we conducted ablations with two alternative grouping strategies. The first strategy uses farthest point sampling (FPS), which is commonly adopted in point cloud models, to select anchor points followed by clustering. The second strategy employs spatial-color k-means clustering, where both the color coefficients and the positional information of the Gaussians are equally weighted (0.5) when forming groups. The experimental results are reported in the new ablation table in Appendix G.5.
>
> |        | $IS↑$    | $C_{sim}↑$    | $C_{con}↑$  | $IAA↑$|
> |------| :------: | :------: | :------: | :------: |
> | Random sampling  | **4.24**| **0.227**  | **0.881**  | **5.27**  |
> | Farthest point sampling  | 4.05 | 0.196  |0.873  |5.20  |
> | Spatial-color k-means  | 4.21 | **0.227**  |0.879 |5.22  |
>
> We observe that the FPS-based grouping yields the worst performance, while the other two strategies perform comparably. We hypothesize that this is mainly because, in 3DGS scenes, the Gaussian primitives are typically denser in the central regions with complex details and sparser near the scene boundaries. Due to its distance-aware mechanism, the FPS strategy tends to favor sampling anchor points along the global contour of the scene. As a result, sparse boundary primitives are frequently selected as anchors, leading to a relatively fixed and non-uniform distribution of anchor points. In contrast, uniform (random) sampling does not consider spatial distance and therefore more frequently selects anchor points from the dense central areas of the scene, resulting in a more diverse distribution of anchor points. In addition, compared with spatial-color k-means clustering, randomly sampling anchor points achieves similar performance while requiring lower computational cost.
>
> **(2) Loss function.**
>
> As discussed in Sec. 4.6 of the paper, we do not optimize the two objectives simultaneously; instead, we use them separately in different training settings. For this reason, we did not further perform ablations on the relative weighting between the loss terms.
>
> ---
>
> **W3. The paper lacks precise runtime measurements or a performance comparison against other editing methods.**
>
> **Response to W3:** We have added a timing comparison in Appendix M of the revised manuscript. Since all existing baselines perform optimization on a per-scene basis, we report both the amortized editing time on the training set and the editing time on the test set separately for a comprehensive comparison, as shown in the table below.
>
> |   Time (s)   | Training Set  | Test Set  |
> |------| :------: | :------: |
> | I-gs2gs  | 275| 275  |
> | GaussianEditor  |463 | 463  |
> | DGE | 210 | 210  |
> | VF-Editor-M  | 216 | **0.3**  |

---

> > ### Comment · Reviewer_giZT · 2025-11-27
> > **Maintaining positive towards this paper**
> >
> > Thanks for the efforts on additional experiments and explanations. My concerns are addressed and I do not have further questions.

---

> > > ### Author Response · Authors · 2025-11-27
> > >
> > > We sincerely appreciate your review and your positive feedback on our work!

---

> ### Author Response · Authors · 2025-11-22
> **Response to Questions**
>
> **Q1. Many details are missing like dimensions, size, exact steps, what noise model is used, more details on losses, etc.**
>
> **Response to Q1:** Thank you for this valuable suggestion. In the revised manuscript, we have added more implementation details in Appendix D, summarized as follows:
>
> For the Variation Field Generation Module $\mathcal{M}$, we set the number of attention heads to 16, with a head dimension of 64. For the Iterative Parallel Decoding Function $\mathcal{F}$, we use a single attention head with a head dimension of 64. The output dimensions of $\mathcal{F}_1$ and $\mathcal{F}_2$ are set to 3 and 11, respectively. In the collected triplets, the noise has a shape of $1\*4\*64\*64$, and the image has a shape of $1\*3\*512\*512$. To strike a balance between efficiency and visual quality, we set the number of sampling steps in the diffusion model to 50.
>
> During data collection, we employed multiple diffusion models, including IP2P, CtrlColor, and Stable Diffusion 2.1 (Inversion). When computing $\mathcal{L}_{sds}$, we set *guidance\_scale* and *condition\_scale* to 6.5 and 3.5, respectively, and linearly decay the time step $t$ from 800 to 100 over the course of training.
>
> ---
>
> **Q2. What are the failure modes?**
>
> **Response to Q2:** VF-Editor may fail when handling instructions that require populating a scene with a large number of objects from scratch, primarily because it is difficult to approximate many previously non-existent objects solely by transforming existing primitives.
>
> ---
>
> **Q3. More elaboration on the fundamental difference with prior works that utilise 2D diffusion priors on 3D editing tasks.**
>
> **Response to Q3:** Prior works exploit 2D diffusion priors only indirectly, typically by reconstructing and projecting the outputs of 2D diffusion models into 3D space. The black-box nature of neural networks inherently makes them prone to multi-view inconsistency, and the per-scene optimization paradigm further limits flexibility. By redefining the 3DGS editing task as a feed-forward variation prediction problem, we fundamentally avoid multi-view inconsistency and greatly enhance the flexibility of editing.

---

### Official Review · Reviewer_jMsT · 2025-10-29

**Soundness:** 3
**Presentation:** 3
**Contribution:** 3
**Rating:** 8
**Confidence:** 2

**Summary:**

The paper introduces VF-Editor, a feed-forward 3D Gaussian Splatting editor that predicts per-primitive variations (position, scale, opacity, color, rotation) to apply edits directly in 3D rather than via multi-view 2D edits and back-projection. It distills knowledge from 2D editors into a variation predictor composed of a transformer-based variation field generator and two iterative parallel decoding functions. This enables fast editing and fine grained control. Most crucially, the method targets reduced cross-view inconsistencies, flexible composition of edits (called by the authors "variation fusion"), and real-time performance. Limitations include ood generalization gaps and occasional local artifacts when relocating primitives.

**Strengths:**

The main strength of the paper is a neat problem reformulation: instead of predicting edited Gaussians outright, the proposed pipeline predicts per-primitive variations and composes them with the source. This gives a controllable, native 3D editing interface and sidesteps multi-view back-projection issues. Such a framing, together with the random tokenizer and the iterative, parallel decoders for position versus other attributes, feels fresh within 3DGS editing and is well-motivated by the representation's explicit structure.

I am no expert in the area, but the technical design looks coherent and lightweight to me, and the training recipe seems well executed to distill multi-source 2D priors into a single feed-forward predictor.

The evaluation is broad and includes competitive quantitative results, as well as targeted ablations that isolate the benefits of iterative decoding and the parallel decoder. There are clear demonstrations of flexible control like strength scaling, locality, and variation fusion. I would have liked to see even more impressive results, but the ones shown in the demo compare very well to prior art nonetheless. The reported runtime (sub-second) and linear decoding complexity with respect to primitive count are also practical positives for interactive use.

In terms of clarity, the paper does a good job with pipeline schematics and equations making the approach easy to follow.

Overall, to the extent of my expertise and knowledge in the field, I think the paper is a meaningful step that removes a common source of inconsistency in indirect pipelines and could influence further research on 3DGS.

**Weaknesses:**

I think there are a couple areas for improvement that are worth discussing. These cluster around data coverage, evaluation, and metodology.

- The training data is well assembled but perhaps is still small and skewed toward objects, with only a handful of scenes; admittedly the authors note lack of ood support (e.g. new categories or environments), which constrains claims of universality and open-vocabulary editing. A more convincing path may add diverse indoor/outdoor scenes, articulated humans, and CAD-like geometry, and report generalization beyond the curated distributions and six scenes used for training and evaluation. Expanding the test suite and including open-vocabulary or paraphrased instructions would make the generalization story more robust than the current check limited to objects.

- On evaluation, the work relies mainly on CLIP-direction metrics, IS, and an aesthetics score, plus qualitative examples; these are useful but weak proxies for 3D editing fidelity and structural preservation, and can "over-reward" appearance changes that do not respect geometry or multi-view consistency. Adding 3D-aware measures (e.g. normal/geometry preservation for non-geometric edits and multi-view consistency checks) would better support the native 3D claim. A small user study focused on edit correctness and obstruction of original content would complement automated metrics.

- Methodologically, I suggest running a deeper ablation. The random tokenizer and group size choices could introduce sampling variance across runs or scenes; alternative tokenization and sensitivity to Gaussian count would make the lightweight/linear story more compelling.

Finally, some limitations are acknowledged and suggest concrete next steps. Relocating primitives can nudge nearby regions, which hints at limited spatial disentanglement; introducing local attention fields could mitigate this. Overall, the core idea looks promising, but the paper would be stronger with broader scene coverage and out-of-domain tests, 3D-aware metrics and user validation; these seem feasible within the current framework and align with the stated goals.

**Questions:**

- If possible, please include a small user study focused on edit correctness and preservation of untouched regions. This would address proxy-metric brittleness and could change the confidence in the method's real-world reliability.

- Can you maybe expand on generalization beyond the current object-centric distribution and clarify scalability limits? A concrete plan or preliminary results on more diverse scenes (indoor/outdoor, cluttered layouts, articulated subjects) would better support open-vocabulary, scene-level editing. Relatedly, how sensitive is performance to the tokenizer's group size and to Gaussian count, and do you foresee adding or removing primitives during editing to reduce spatial entanglement when relocating content?

---

> ### Author Response · Authors · 2025-11-22
> **Response to W1 and W2**
>
> **W1. The training data is well assembled but perhaps is still small and skewed toward objects, with only a handful of scenes.**
>
> **Response to W1:** Thank you very much for your valuable comments. To expand the amount of 3D scene data within the limited rebuttal time, instead of collecting additional real-world scenes, we generated 115 outdoor street-view scenes using LucidDreamer [1]. We conducted new experiments on this extended dataset and have updated Table 1, Table 2, Figure 7, Table 4, as well as the corresponding descriptions in the paper. As shown in Figure 7, our method demonstrates reasonable generalization ability even on previously unseen scenes.
>
> In future work, we plan to collect more diverse types of data. However, our current model is specifically designed for 3DGS-style data; extending it to articulated humans or CAD-like geometry would require substantial architectural redesign. We must acknowledge that such a large-scale modification is difficult to accomplish during the rebuttal period. Nonetheless, with an appropriate encoder–decoder, the core idea of our method can, in principle, transfer to any data format with explicit structure.
>
> Additionally, we have included more editing results under paraphrased instructions in Appendix I and provided further analysis.
>
> [1] Chung J, Lee S, Nam H, et al. Luciddreamer: Domain-free generation of 3d gaussian splatting scenes[J]. arXiv preprint arXiv:2311.13384, 2023.
>
> ___
>
> **W2. Adding 3D-aware measures (e.g. normal/geometry preservation for non-geometric edits and multi-view consistency checks) would better support the native 3D claim. A small user study focused on edit correctness and obstruction of original content would complement automated metrics.**
>
> **Response to W2:** Thank you for this valuable suggestion. We fully agree that 3D fidelity and consistency require more robust evaluation. To address this, we have added three complementary analyses in Appendix L: 3D geometry preservation, multi-view consistency checks, and user study results. We believe these additions significantly strengthen the credibility of our claims.
>
> - 3D Geometry Preservation. To assess whether our edits affect the underlying 3D structure for non-geometric edits, we simplify the 3DGS representation into point clouds and compute the Chamfer Distance and F-score (τ = 0.01) between the point clouds before and after applying two representative non-geometric edits (“*make its color look like rainbow*” and “*make its color look like gold*”). Across 100 trials, the average results are: Chamfer Distance = 2.4273e-05, F-score = 0.9730. These outcomes indicate that the geometry of the 3D model remains almost entirely intact.
>
> - Multi-view Consistency Checks. The metric $C_{con}$ in Table 2 is a widely used indicator of multi-view consistency in the 3DGS editing literature. To provide a more comprehensive assessment of view consistency, we introduce a new metric called $GPT-score_{edit}$. We render some edited views and feed the resulting images into GPT-5, using a carefully designed prompt to score their cross-view consistency. The average scores over the 11 test cases are reported in the table below. The full prompt is provided in Appendix L.
>
> |   | $GPT-score_{edit}$ |
> | ------ | :------: |
> | I-gs2gs  | 6.3  |
> | GaussianEditor  | 6.7  |
> | DGE  | 7.5  |
> | VF-Editor-M  | **8.3**  |
>
>
> - User Study. We also conducted a user study using a questionnaire-based evaluation. For each question, volunteers were shown videos rendered from the edited results of different methods, along with the corresponding editing instruction, all in randomized order. Participants rated each result in terms of Aesthetic Quality, Instruction Following, and Faithfulness to the Original 3D Model, using a scale from 0 (worst) to 10 (best). We collected scores from 11 volunteers across 5 sets of data, and the averaged results are reported in the table.
>
> |   | Aesthetic Quality | Instruction Following|Faithfulness |
> | ------ | :------: | :------: | :------: |
> | I-gs2gs  | 4.7  |5.1 | 6.4|
> | GaussianEditor  |5.9  |6.0 |5.9 |
> | DGE  | 6.4 | 5.9| 7.5|
> | VF-Editor-M  | **9.3**  | **9.0**| **9.0**|

---

> ### Author Response · Authors · 2025-11-22
> **Response to W3**
>
> **W3. Methodologically, I suggest running a deeper ablation.**
>
> **Response to W3:** Thank you very much for your constructive suggestions. In the revised version of our paper, we have added two additional ablation studies on the GObj subset in the Appendix G.5.
>
> **(1) Changing the grouping strategy in the tokenizer.**
>
> We implemented two new grouping methods: (i) selecting anchor points using farthest point sampling (FPS), a technique commonly used in point-cloud models, followed by clustering; and (ii) spatial–color k-means clustering, where both the color coefficients and the positions of the Gaussians are equally weighted (0.5). The experimental results are shown in the table below. We observe that FPS leads to the weakest performance, while the remaining two methods yield comparable results.
> We hypothesize that this is mainly because in 3DGS scenes, the Gaussian primitives are typically denser in central regions with complex structures and sparser near scene boundaries. Due to its distance-aware mechanism, FPS tends to select anchor points along the scene’s global contour. As a result, sparse boundary primitives are frequently chosen as anchors, leading to a relatively fixed and uneven distribution of anchor points. In contrast, uniform (random) sampling does not consider spatial distances and therefore selects anchors more frequently from the dense central regions, resulting in a more diverse distribution of anchor points. In addition, although the performance of random sampling and spatial–color k-means clustering is similar, random sampling followed by clustering requires substantially less computation.
>
> |        | $IS↑$    | $C_{sim}↑$    | $C_{con}↑$  | $IAA↑$|
> |------| :------: | :------: | :------: | :------: |
> | Random sampling  | **4.24**| **0.227**  | **0.881**  | **5.27**  |
> | Farthest point sampling  | 4.05 | 0.196  |0.873  |5.20  |
> | Spatial-color k-means  | 4.21 | **0.227**  |0.879 |5.22  |
>
>
> **(2) Further analysis of group size.**
>
> We then fixed the anchor sampling strategy to random sampling and varied the group size (adjusting the number of groups proportionally) to examine its influence on model performance. As shown in the table, within a reasonable range, different group sizes have only a minor impact on performance.
>
> |   Group size   | $IS↑$    | $C_{sim}↑$    | $C_{con}↑$  | $IAA↑$|
> |------| :------: | :------: | :------: | :------: |
> | 64  | **4.24**| **0.228**  | 0.879 | **5.27**  |
> | 128  | **4.24** | 0.227  | **0.881**  |**5.27**  |
> | 192  | 4.22 | 0.223  |0.876 |5.24  |
>
> We hope that these results can address your concerns regarding our random tokenizer.
>
> ---
>
> **W. Finally, some limitations are acknowledged and suggest concrete next steps.**
>
> **Response:** (1) We sincerely appreciate your valuable suggestion. Introducing local attention fields is indeed an excellent idea, and we plan to explore incorporating this direction into our algorithm in future iterations.
> (2) In the revised manuscript, we have added evaluations covering a broader range of scenes, and we further analyze boundary cases of the out-of-domain tests in Appendix I.
> (3) We have also included additional results on 3D-aware metrics and a user study in Appendix L.

---

> > ### Comment · Reviewer_jMsT · 2025-11-27
> >
> > Thank you for your detailed answers and additional results. I have no further concerns, I confirm my original recommendation to accept the paper to the conference.

---

> > > ### Author Response · Authors · 2025-11-28
> > >
> > > Thank you again for your valuable comments and for your recognition of our work.

---

> ### Author Response · Authors · 2025-11-22
> **Response to Q1 and Q2**
>
> **Q1. If possible, please include a small user study focused on edit correctness and preservation of untouched regions.**
>
> **Response to Q1:** Thank you for your suggestion. Please refer to the **Response to W2** for the relevant data and our detailed reply.
>
> ---
>
> **Q2. Can you maybe expand on generalization beyond the current object-centric distribution and clarify scalability limits? A concrete plan or preliminary results on more diverse scenes (indoor/outdoor, cluttered layouts, articulated subjects) would better support open-vocabulary, scene-level editing. Relatedly, how sensitive is performance to the tokenizer's group size and to Gaussian count, and do you foresee adding or removing primitives during editing to reduce spatial entanglement when relocating content?**
>
> **Response to Q2:**
>
> **(1) Expand on generalization beyond the current object-centric distribution and clarify scalability limits.**
>
> In the revised manuscript, we augmented our training data using 3D street-scene samples generated by LucidDreamer. Since our work is the first to introduce variation-aware 3D Gaussian editing, this paper primarily focuses on validating the feasibility of this new perspective. In future work, we plan to enlarge both the dataset and the model scale to develop a more powerful system. At the current stage, the main limitations come from computational resources and the inherent capacity of existing 2D editors.
>
> **(2) How sensitive is performance to the tokenizer’s group size and to Gaussian count.**
>
> As discussed in the **Response to W3**, within a reasonable range, changing the group size has only a limited impact on model performance. This is mainly because the self-attention layers in the Variation Field Generative Module naturally aggregate global information. As long as the grouping remains within a sensible range, the model can adaptively capture the global structure of objects composed of varying numbers of Gaussian primitives.
>
> **(3) Adding or removing primitives during editing to reduce spatial entanglement when relocating content.**
>
> We sincerely appreciate your insightful comment. We have begun experimenting with mechanisms such as a “generation branch” or a “reset head” to add or remove primitives during editing. However, our preliminary results indicate that training becomes less stable under these operations. We suspect this is due to the fact that directly generating or deleting primitives substantially enlarges the solution space of the editing problem, making optimization more challenging.

---

### Official Review · Reviewer_rKQ5 · 2025-11-01

**Soundness:** 3
**Presentation:** 3
**Contribution:** 3
**Rating:** 6
**Confidence:** 4

**Summary:**

This paper presents VF-Editor, a novel framework for "native" 3D Gaussian Splatting (3DGS) editing. The core idea of VF-Editor is to reframe 3D editing as a feed-forward variation prediction problem. Instead of predicting a new 3D scene, the method learns a single "variation predictor" ($\mathcal{P}_{\theta}$) that directly outputs the change (or variation, $\Delta$) for all attributes of each 3D Gaussian based on a text instruction.

This predictor is trained by distilling knowledge from 2D editors. The training process collects {initial noise}-{instruction}-{edited 2D image} triplets. The resulting system, VF-Editor, can perform 3D edits in ~0.3 seconds.

**Strengths:**

Predicting changes (Δ) instead of the final result is a smart and natural fit for 3D Gaussian Splatting. Since 3DGS is made up of explicit, editable primitives, it makes more sense to directly modificate their parameters rather than trying to infer 3D edits indirectly from 2D images.

The feed-forward nature provides a significant speed-up (0.3s) over iterative optimization methods.

**Weaknesses:**

Data Dependency: The entire framework is built on offline triplet collection ($\mathcal{L}_{din}$). Table 1 indicates that 28,932 triplets were required for only 20 instructions. This approach seems to scale very poorly for a truly "open-vocabulary" editor. The paper admits in Sec. 4.6 that it does not support "out-of-domain editing" without fine-tuning (Fig. 14). This suggests the model is learning a mapping for a fixed set of instructions, not a general-purpose, compositional understanding of language.


Limited Generalization: The "unseen instruction" results in Fig. 7 ("Apply clown makeup") are semantically almost identical to a training instruction ("Turn him into a clown"). This feels less like true zero-shot generalization and more like interpolation within a learned semantic space.

Supervision Quality: The training relies on 2D editors. DDIM is not perfect. If the 2D editor (e.g., IP2P) produces a low-quality or inconsistent edit for a given view, this flawed supervision is baked into the triplet and, by extension, the 3D model. The paper doesn't discuss how it handles or filters "bad" 2D edits during the triplet collection phase.

**Questions:**

Since $\mathcal{L}_{din}$ trains on single-view edits, how does the model handle conflicting 2D edits for the same 3D-instruction pair during training? For example, if IP2P produces a red hat from the front view but a blue hat from the side view for the "put on a party hat" instruction, how is the model learn under such data?

---

> ### Author Response · Authors · 2025-11-22
> **Response to W1 and W2**
>
> **W1. Data Dependency**
>
> **Response to W1:** We sincerely thank you for the thoughtful feedback. There are two main reasons why our training process uses a relatively large number of triplets.
>
> - Our model is trained entirely from scratch, and using too little data easily leads to severe overfitting.
> - To demonstrate that our approach can support a wide variety of editing types, many of the editing instructions in our dataset correspond to fundamentally different categories of edits. This diversity requires the model to learn multiple independent forms of knowledge, which in turn increases the need for training data.
>
> That said, it is foreseeable that once the model has learned a particular category of edit, acquiring unseen concepts within the same category should become substantially easier. To verify this intuition, we conducted a simple experiment. We introduced a new instruction, “*make its color bright turquoise*”, containing the unseen concept “**bright turquoise**”. Following the procedure described in Section 3.2.2, we collected only 25 triplets for this new instruction and fine-tuned the model. As shown in Figure 17, the model successfully learned the new concept with ease. This demonstrates that for edits within the same category, even novel concepts can be learned with very few training examples. For instance, once the model has learned “*make its color look like gold*,” learning “*make its color bright turquoise*” becomes much easier. We have added this experiment to Appendix K.
>
> Finally, we would like to emphasize that our training data is primarily collected through an automated pipeline, so the human labor cost involved in the overall process is quite low.
>
> ---
>
>   **W2. Limited Generalization**
>
> **Response to W2:**  Please allow us to respond to this concern from three perspectives: comparison with baselines, comparison with foundation models in other domains, and the primary objective of our work.
>
> - **Comparison with baselines.** Compared with existing 3DGS editing methods, our approach demonstrates noticeably stronger generalization ability. Due to their reliance on per-scene optimization and the absence of any mechanism for storing editable knowledge, current baselines inherently lack zero-shot editing capabilities.
>
> - **Comparison with foundation models in other fields.** We sincerely acknowledge that the current version of our 3D editing model is still less generalizable than large 3D generative foundation models or 2D editing foundation models. Fundamentally, the generalization ability of neural networks relies on relevant knowledge, which in turn requires larger datasets, larger model capacities, and substantially more training resources. Due to practical constraints, we are currently unable to collect data at the same scale as those foundation models or train a model of comparable size.
>
> - **Objective of this work.** As stated in the Abstract and Introduction, the primary goal of VF-Editor is to enhance the flexibility and consistency of 3DGS editing by introducing a new editing paradigm. Our experiments demonstrate that VF-Editor achieves clear advantages in both aspects. As an additional property, although the current generalization ability remains limited to the in-domain setting, we believe that this first step toward enabling generalization in 3DGS editing can provide a new perspective and inspire future research in this area.

---

> ### Author Response · Authors · 2025-11-22
> **Response to W3**
>
> **W3. Supervision Quality**
>
> **Response to W3:** In our method, the impact of low-quality 2D edited images and that of inconsistent edited images are fundamentally different, which leads us to treat them differently.
>
> - First, regarding low-quality 2D edited images, we fully agree with your observation that such images can indeed degrade the final editing results. In the current version, we apply a simple filtering step using an image aesthetic scorer [1]. We discard triplets whose edited view receives a low score, which typically corresponds to clear 2D failures such as heavy artifacts, severely distorted structures, or visually collapsed edits. If higher aesthetic fidelity is desired, we believe incorporating a multimodal large model for evaluation could potentially yield more accurate filtering results. Thank you for pointing out this issue — we have added the corresponding clarifications in Appendix D of the revised manuscript.
>
> - Second, **regarding inconsistent edited results, we do not perform any filtering, because our algorithm is inherently capable of handling such variations**. As described in line 203 of the paper, *our method accommodates diverse edited outcomes by storing rather than restricting probability flow*, as illustrated in Figure 1 for the colorization task (and further demonstrated in our demo.mp4). We noticed that the concern you raised in your question is closely related to this mechanism. Therefore, please allow us to provide a more detailed explanation in the “**Response to Q1**.”
>
> [1] Yi R, Tian H, Gu Z, et al. Towards artistic image aesthetics assessment: a large-scale dataset and a new method[C]//Proceedings of the IEEE/CVF Conference on Computer Vision and Pattern Recognition. 2023: 22388-22397.

---

> ### Author Response · Authors · 2025-11-22
> **Response to Q1**
>
> **Q1. Since trains on single-view edits, how does the model handle conflicting 2D edits for the same 3D-instruction pair during training?**
>
> **Response to Q1:** Thank you very much for raising this important question — the answer indeed touches on a key insight behind our method. Although single-view 2D edited images in the training data may contain inconsistent outcomes across different viewpoints (e.g., a red hat in the front view but a blue hat in the side view), the training and inference mechanisms of VF-Editor ensure that the model is not misled by such conflicting supervision. Instead, it learns to produce cross-view consistent 3D edits.
>
> The reason is that **the model does not attempt to memorize a fixed answer for each specific viewpoint within a training triplet.** Rather, it learns from two complementary sources of information: (1) explicit information — one plausible 2D edited solution for the current viewpoint generated by the editor; and
> (2) **more importantly, implicit information — the underlying instruction editing pattern**, i.e., how the instruction should transform the object’s global structure or appearance in 3D space. As noted in line 298, “*We observe that, given sufficient training data, $\mathcal{P}_{\theta}$ is capable of inferring changes in novel views based on variations observed in the known ones.*” This ability emerges because the model abstracts an instruction-level editing pattern rather than overfitting to individual views.
>
> Importantly, this cross-view consistency does not rely on multi-view–consistent 2D supervision. Instead, it is enabled by our probability-flow–preserving mechanism, described in lines 197–210. In our Variation Field Generation Module, the “key noise” from the 2D editing process is also included as part of the model input. This design allows the distilled model to learn a distribution rather than a single deterministic solution. During inference, users simply sample noise from a standard Gaussian distribution—just as in diffusion models—and VF-Editor generates an edited result drawn from this learned, plausible distribution. This idea of using noise to retain probability flow has also proven effective in diffusion-acceleration literature [2].
>
> This mechanism is strongly evidenced by our colorization experiments, where the space of valid solutions is extremely diverse and achieving multi-view-consistent 2D edits is nearly impossible. Yet, VF-Editor still produces cross-view-consistent 3D colorization. For example, in the first colorization case in Figure 1, **the training set does not contain 2D edited images showing white petals, brown stamens, and yellow leaves simultaneously across all views—a combination that is almost impossible for independent 2D edits to produce.** **Nevertheless, the model generates structurally consistent results for every viewpoint.** This indicates that the model actually learns a stable 3D transformation pattern induced by the instruction, rather than replicating per-view samples seen during training.
>
> Therefore, what appear to be “conflicting” 2D edits do not harm the training process. On the contrary, they provide diverse samples of the underlying instruction editing pattern, enabling the model to better distill the stable, shared 3D transformation factors across views. The ability of VF-Editor to achieve multi-view-consistent editing from single-view supervision ultimately stems from this principle of learning a distribution instead of a fixed solution, together with the model’s capacity to abstract instruction-level editing patterns.
>
> [2] Kang M, Zhang R, Barnes C, et al. Distilling diffusion models into conditional gans[C]//European Conference on Computer Vision. Cham: Springer Nature Switzerland, 2024: 428-447.

---

### Official Review · Reviewer_Ze3B · 2025-11-02

**Soundness:** 2
**Presentation:** 2
**Contribution:** 2
**Rating:** 4
**Confidence:** 4

**Summary:**

The paper presents VF-Editor, a native, feedforward approach for direct 3D Gaussian Splatting (3DGS) editing by distilling multi-source 2D editing knowledge into a unified variation predictor network. Instead of using indirect, view-by-view projection-and-reconstruction pipelines prone to cross-view inconsistencies, VF-Editor predicts per-primitive attribute variations (position, scale, opacity, color, rotation) using learned features and incorporates probabilistic flows. The authors introduce a variation field generation module and parallel decoding functions, with extensive experiments demonstrating improved flexibility, consistency, and real-time operation across editing tasks. The paper also provides ablation studies and assesses generalization to unseen data and instructions.

**Strengths:**

1. The paper responds to a well-recognized limitation in 3DGS editing: the cross-view inconsistencies inherent to indirect, 2D-edit-then-project pipelines. The authors' framing is accurate and well-motivated, making a convincing case for the need for direct, native 3D editing.
2. The feedforward variation predictor architecture is novel. The random tokenizer, transformer-based variation field generator, and parallel iterative decoding functions offer a clear path to efficient, scalable editing. The distinction between mean and other attributes in decoding further reflects architectural care to handle attribute coupling.
3. VF-Editor's distillation pipeline can combine knowledge from diverse 2D editors, such as IP2P for DDIM-based edits and CtrlColor for style/color tasks, with training informed by different datasets. This unification of editing capabilities is practically impactful.
4. The experiments cover a range of editing instructions, including both qualitative and quantitative reporting (Table 2, Table 4), and address both baseline comparisons and ablations. Figure 3 demonstrates per-instruction failure modes for baselines, while Figures 4–6 visualize the contribution of each architectural choice and VF-Editor's flexibility. Figure 6, in particular, makes the interpretability and composability of the approach tangible, which is a strong point.

**Weaknesses:**

1. Although the Related Work section is relatively comprehensive for 3DGS and 2D distillation methods, several highly pertinent and recent methods are missing. In particular:
    - 3DSceneEditor (Yan et al., 2024) is another fully 3D-based native editing pipeline leveraging Gaussian Splatting. This work should be directly compared with or discussed in Section 2 and as a baseline in Section 4.2/Table 2.
    - Gaussian Splatting in Style (Saroha et al., 2024), which introduces neural style transfer techniques directly into the 3DGS domain, offering valuable insights for stylization and attribute manipulation. This work should at least be discussed contextually.
    - ManiGaussian (Lu et al., 2024) and Depth-Regularized Optimization for 3D Gaussian Splatting (Chung et al., 2024) are also relevant for scene dynamics and editing accuracy. Their omission affects the completeness and currentness of the empirical/positional claims, suggesting the need for updated discussion and possibly additional comparative experimentation.
2. The claims about generalization (Section 4.5, Table 4, Figure 7) are largely limited to within-domain samples or instructions that are semantically related to those seen during training. There is explicit acknowledgment (Line 475)  that the model "does not yet support out-of-domain editing" but no serious attempt is made to systematically characterize the failure modes or boundary conditions of generalization. This diminishes the universality and practical significance of the claimed contributions.
3. While the structure of the predictor and the use of zero-init linear layers are explained, the choice of the specific architecture (transformer blocks with and without self-attention) and the separation of decoding for position mean vs. attributes is mostly justified by empirical ablation rather than solid theoretical backing. For instance, Section 3.1 references attribute "intercoupling" but does not model it explicitly or provide a concrete analysis of when/why the architecture remains robust to different types of edits. It would strengthen the work to either formalize these choices further or relate them to established results in representation learning or signal decomposition.

**Questions:**

See the questions listed above.

---

> ### Author Response · Authors · 2025-11-22
> **Response to W1 and W2**
>
> **W1. Several highly pertinent and recent methods are missing.**
>
> **Response to W1:** Thank you very much for providing these related works. We have carefully incorporated them into the revised manuscript, discussing or citing them in both the Introduction and Related Work sections. Specifically:
>
> - In Section 2, we discuss 3DSceneEditor [1] and GSS [2]: *3DSceneEditor achieves object addition, deletion, or relocation by segmenting individual objects from the scene, while GSS realizes 3D stylization by modifying the color coefficients of each primitive. However, since each of them supports only a single type of edit, they are insufficient for more flexible editing scenarios.*
>
> - In the Introduction, we cite ManiGaussian [3] and DRGS [4] to further strengthen the completeness of our literature review.
>
> However, these works cannot be directly compared with ours at the experimental level. Please allow us to explain this in detail:
>
> - 3DSceneEditor [1] and GSS [2] are limited to specific types of editing tasks, which **differ fundamentally from the problem setting of our work**. 3DSceneEditor relies on a two-stage “detect–edit’’ pipeline to heuristically manipulate objects (e.g., moving or removing them), while GSS focuses on transferring color style from a reference image to 3D Gaussians. Neither framework can handle the shape–style transformation tasks considered in our paper, such as “*Turn him into the Tolkien Elf.*” In addition, **the code for both papers is not publicly available**, making a fair empirical comparison currently infeasible.
>
> - ManiGaussian [3] and DRGS [4], although related to 3D Gaussian Splatting, **focus on tasks that are substantially different from ours**. ManiGaussian investigates multi-task robot manipulation under language instructions, while DRGS focuses on sparse-view reconstruction. Their methods do not appear directly applicable to 3D editing tasks
>
> [1] Yan Z, Li L, Shao Y, et al. 3dsceneeditor: Controllable 3d scene editing with gaussian splatting[J]. arXiv preprint arXiv:2412.01583, 2024.
>
> [2] Saroha A, Gladkova M, Curreli C, et al. Gaussian splatting in style[C]//DAGM German Conference on Pattern Recognition. Cham: Springer Nature Switzerland, 2024: 234-251.
>
> [3] Lu G, Zhang S, Wang Z, et al. Manigaussian: Dynamic gaussian splatting for multi-task robotic manipulation[C]//European Conference on Computer Vision. Cham: Springer Nature Switzerland, 2024: 349-366.
>
> [4] Chung J, Oh J, Lee K M. Depth-regularized optimization for 3d gaussian splatting in few-shot images[C]//Proceedings of the IEEE/CVF Conference on Computer Vision and Pattern Recognition. 2024: 811-820.
>
> ---
>
> **W2. No serious attempt is made to systematically characterize the failure modes or boundary conditions of generalization.**
>
> **Response to W2:** Thank you for your valuable comments. We have added an analysis of the boundary conditions of generalization in Appendix I of the revised manuscript, which includes two parts: (1) fixing the instruction while changing the edited 3D model, and (2) fixing the 3D model while varying the semantics of the instruction. From Figures 15 and 16, we observe the following:
>
> (1) For general-purpose instructions, the model is able to produce reasonable editing results even on out-of-domain data. (2) As the test instruction gradually deviates semantically from the original one (e.g., “*make its color look like a rainbow*”), the model’s ability to follow the instruction also degrades. In Figure 16, when the test instruction remains semantically related to the training instructions—such as “*apply a vivid spectrum of colors*”—the editing outputs remain reasonable and exhibit a distribution distinct from the outputs obtained using the original training instruction. However, when the semantics of the test instruction differ substantially from those seen during training—for instance, “*make the lighting dramatic and moody*”—the model’s behavior begins to diverge from the instruction. We suspect this occurs because the model lacks the ability to generalize to entirely unseen semantic concepts that were never learned during training.

---

> ### Author Response · Authors · 2025-11-22
> **Response to W3**
>
> **W3. The choice of the specific architecture (transformer blocks with and without self-attention) and the separation of decoding for position mean vs. attributes is mostly justified by empirical ablation rather than solid theoretical backing.**
>
> **Response to W3:** We sincerely thank you for pointing out the missing explanation. We have added a more detailed theoretical discussion in Appendix J, and summarize the core ideas below:
>
> **(1) Why use transformer blocks _with_ self-attention in the Variation Field Generation Module ($\mathcal{M}$), but _without_ self-attention in the Iterative Parallel Decoding Function ($\mathcal{F}$)?**
>
> Our Variation Predictor can be viewed as a _structured factorization_ of the 3D editing operator:
>
> - $\mathcal{M}$ is responsible for extracting global correlations from the input scene and generating a _global variation field_.
> - $\mathcal{F}$ then performs lightweight _per-primitive_ update decoding for each Gaussian primitive.
>
> This factorization is analogous to the common encoder–latent–decoder structure in representation learning: the variation field $f_{\Delta}$ serves as a global, low-dimensional, cross-primitive _latent basis_, while $\mathcal{F}_1 / \mathcal{F}_2$ are only responsible for conditional decoding (conditional readout).
>
> Under this design:
>
> - **Why $\mathcal{M}$ needs self-attention.** The variation field must aggregate structural information from the _entire_ scene. Given the presence of a tokenizer, self-attention is the most effective global information aggregation mechanism, as it can capture cross-primitive correlations with a controllable computational budget.
>
> - **Why $\mathcal{F}$ intentionally avoids self-attention.** Once a global variation field is available, introducing self-attention inside $\mathcal{F}$ would force all Gaussian primitives to interact pairwise. This would increase the complexity from $O(N)$ to $O(N^2)$, which becomes prohibitively expensive for large-scale scenes. At the same time, we would like each primitive's update to remain _conditionally independent_ given the global field so that decoding can be fully parallelized. Therefore, $\mathcal{F}$ uses only cross-attention, conditioning on the same shared variation field. This keeps the decoding complexity linear in $N$, which is crucial for large scenes.
>
> **(2) Why do we separate $\mathcal{F}_1$ (mean) from $\mathcal{F}_2$ (scale/opacity/color/rotation)?**
>
> This design is motivated by the analytical structure of the 3DGS render (Eq. (7)–(8) in App.B.1). The rendered color $C = f(\mu, s, \alpha, c, r)$ admits the following Jacobian under a first-order Taylor expansion with respect to the parameters:
>
> $$
> J_{\theta} =
> \begin{bmatrix}
> J_{\mu\mu} & J_{\mu A} \\\\
> J_{A\mu} & J_{AA}
> \end{bmatrix}, \quad A = (s, \alpha, c, r).
> $$
>
> The non-zero off-diagonal blocks indicate that the geometric position $\mu$ and the appearance attributes $A$ are _intercoupled_:
>
> - Changing $\mu$ alters the distribution of primitives in screen space, which in turn affects the effective contribution of their opacity/scale/color.
> - Changing appearance attributes likewise affects how gradients propagate along geometric directions.
>
> Some 3DGS reconstruction methods [5][6] also adopt similar strategies to avoid letting the appearance parameters (scale, color, opacity) "absorb the error."
>
> Against this background, our $\mathcal{F}_1/\mathcal{F}_2$ design can be interpreted as an _approximate block-coordinate update / block-diagonalization_ of the Jacobian:
>
> - $\mathcal{F}_1$ first predicts the geometric position $\mu$, preventing geometric gradients from being overwhelmed or distorted by appearance parameters.
> - $\mathcal{F}_2$ then updates the appearance attributes after the geometry has been stabilized.
>
> This structure alleviates the optimization instability caused by the strong coupling between geometry and appearance, enabling the system to handle both _geometric edits_ and _appearance edits_ more reliably.
>
> As shown in Figure 4 and Table 3:
>
> - Directly decoding all attributes jointly is prone to failure under displacement-type edits.
> - Our iterative decoding remains stable across almost all types of edits.
>
> [5] Charatan D, Li S L, Tagliasacchi A, et al. pixelsplat: 3d gaussian splats from image pairs for scalable generalizable 3d reconstruction[C]//Proceedings of the IEEE/CVF conference on computer vision and pattern recognition. 2024: 19457-19467.
>
> [6] Lan L, Shao T, Lu Z, et al. 3dgs2: Near second-order converging 3d gaussian splatting[C]//Proceedings of the Special Interest Group on Computer Graphics and Interactive Techniques Conference Conference Papers. 2025: 1-10.

---

### Author Response · Authors · 2025-11-22

Dear Reviewers,

We would like to sincerely thank you for the time and effort you have devoted to reviewing our manuscript, as well as for your insightful and constructive comments. We are pleased that the novelty and soundness of our idea have been recognized.

In accordance with your suggestions, we have revised the manuscript and uploaded an updated version, in which all modified text is highlighted in blue. We will respond to each reviewer's comments individually over the next couple of hours.

Thank you for your time and consideration.

Sincerely,

The Authors

---

### Author Response · Authors · 2025-12-02
**Summary of Rebuttal and Revisions for Paper 6652**

Dear PC, SAC, and AC,

We sincerely appreciate your time and effort in handling our manuscript. We have uploaded the revised manuscript with changes marked in blue and provided detailed responses to each reviewer below. To facilitate your review, we have summarized the reviewers' main points and our corresponding responses here.

**1. All reviewers positively acknowledged the novelty of our core idea and the soundness of our framework design.**
- Reviewer `Ze3B`: *"The authors' framing is accurate and well-motivated... The feedforward variation predictor architecture is novel"*;
- Reviewer `rKQ5`: *"Predicting changes ($\Delta$) instead of the final result is a smart and natural fit for 3D Gaussian Splatting"*;
- Reviewer `jMsT`: *"Such a framing... feels fresh within 3DGS editing and is well-motivated by the representation's explicit structure"*;
- Reviewer `giZT`: *"The core idea... is novel and addresses limitations of existing indirect editing methods"*.

Additionally, the reviewers highlighted the paper's strengths regarding training (Reviewer `Ze3B`), experimentation (Reviewers `Ze3B`, `jMsT`, `giZT`), and efficiency (Reviewers `rKQ5`, `jMsT`, `giZT`). During the previous discussion phase, reviewers `jMsT` and `giZT` explicitly stated that all their concerns had been resolved, while reviewers `Ze3B` and `rKQ5` had not yet participated in the discussion.


**2. Summary of Concerns and Responses**

**(1) Reviewer `Ze3B` (Score: 4)**
- **Missing related work:** We have cited or discussed all the related work mentioned by the reviewer in the revised Introduction and Related Work sections. We also clarified the key differences between our method and these works in our detailed response.
- **Lack of systematic analysis on failure modes/boundary conditions:** In the revised manuscript, we have conducted a systematic analysis of boundary conditions using visualization experiments in App. I.
- **Request for architectural rationale:** We have added the relevant theory behind the key structural designs mentioned by the reviewer in App. J and provided an in-depth analysis.

**(2) Reviewer `rKQ5` (Score: 6)**
* **Use of large numbers of triplets in training:** We explained that the two main reasons for this are avoiding overfitting and learning diverse editing capabilities. In App. K, we demonstrated experimentally that *for edits within the same category, even novel concepts can be learned with very few (25) triplets.*
* **Limited zero-shot generalization:** We responsed to this concern from three aspects. First, **compared to baselines, our approach demonstrates noticeably stronger generalization ability**. Second, we acknowledge that the current version of our 3D editing model is still less generalizable than large 3D generative foundation models or 2D editing foundation models, as achieving similar generalization requires massive data and training resources. Finally, we clarified that the primary goal of VF-Editor is to enhance the flexibility and consistency of 3DGS editing.
* **How to handle low-quality/inconsistent 2D training data:** We employed an aesthetic scorer to filter low-quality images, with relevant details added to App. D. Regarding view-inconsistent 2D data, we did not apply filtering *because our algorithm is inherently capable of handling such variations*. We provided a detailed explanation in our "Response to W3" and "Response to Q1", referencing relevant sections of our paper.

**(3) Reviewer `jMsT` (Score: 8)**
* The reviewer's comments included extending our method to more diverse scenes, adding more metrics, and conducting further ablation studies. We addressed these suggestions point-by-point. In the discussion phase, the reviewer confirmed: **"I have no further concerns."** To save your time, we do not elaborate on the specific details here.

**(4) Reviewer `giZT` (Score: 6)**
* The reviewer's comments included clarifying differences from related works (DiffusioNeRF, In-N-Out), conducting more ablation studies, adding time comparisons, supplementing implementation details, and clarifying failure modes. We responded to each point individually. In the discussion phase, the reviewer stated: **"my concerns are addressed and I do not have further questions."**

**Final Remark**

In conclusion, we have systematically addressed all reviewer concerns and highlighted all significant revisions in the updated manuscript. We believe our rebuttal has effectively resolved the issues raised by the reviewers. We are very grateful for the constructive feedback, which has significantly enhanced the quality and completeness of our work.

Thank you once again for your time and consideration.

Sincerely,

The Authors

---

### Meta-Review · Area_Chair_wz4N · 2026-01-03

**Summary:**

Four reviewers assessed this submission, with initial ratings ranging from borderline reject to acceptance. The primary concerns centered on the method's generalization capabilities, the omission of key related works, and the robustness of the evaluation metrics. Reviewers also questioned the handling of inconsistent or low-quality 2D supervision and sought deeper justifications for architectural choices and hyperparameter sensitivity. The authors provided a comprehensive rebuttal that included additional comparisons, runtime measurements, failure mode analyses, and clarifications on handling conflicting supervision. Given the effective response to these critiques, AC recommends acceptance.

**Reviewer Concerns:**

The consensus concern regarding the method's generalization and the scalability of the data collection pipeline (raised by Reviewers Ze3B, rKQ5, and jMsT) was partially addressed by the authors through clarifications on the scope of open-vocabulary editing and additional experimental results demonstrating performance on unseen instructions. The critiques regarding missing related work and insufficient differentiation from prior art (Reviewers Ze3B and giZT) were resolved by the inclusion of comprehensive discussions on recent methods. Technical concerns about the quality of supervision, specifically how the model handles conflicting or poor-quality 2D edits (Reviewer rKQ5), were answered with detailed explanations of the noise handling and training dynamics. Finally, limitations in the evaluation suite, such as the lack of 3D-aware metrics, runtime stats, and ablation studies (Reviewers jMsT and giZT), were remedied by providing the requested runtime measurements, expanded metrics, and deeper analyses of the tokenizer and architectural components.

**Reviewer Scores:**

Reviewer Ze3B would likely increase his/her score to a positive score (like 6), as the authors successfully integrated the missing related works and provided the requested characterization of generalization boundaries, addressing the reviewer's main grounds for the initial borderline rejection. Reviewer rKQ5 is expected to maintain his/her score or even raise the score to 8, given that his/her specific concerns about data dependency and the handling of conflicting multi-view supervision were effectively countered with logical justifications and evidence. Reviewer jMsT has acknowledged the authors’ rebuttal addressed his/her concerns, and maintained the initial high score of 8. Reviewer giZT has also confirmed that the authors’ rebuttal addressed his/her concerns, so he/she will probably maintain the initial positive score.

---

### Decision · Program_Chairs · 2026-01-26

Accept (Poster)